# Regional-scale modelling for the assessment of atmospheric particulate matter concentrations at rural background locations in Europe

Goran Gašparac[1], Amela Jeričević[2], Prashant Kumar[3], and Grisogono Branko[4]

[1]Croatian Meteorological and Hydrological Service, Zagreb, Croatia
[2]Croatia Control Ltd., Zagreb, Croatia
[3]Global Centre for Clean Air Research (GCARE), Department of Civil and Environmental Engineering, Faculty of Engineering and Physical Sciences, University of Surrey, Guildford GU2 7XH, United Kingdom
[4]Department of Geophysics, Faculty of Science, University of Zagreb, Zagreb, Croatia

**Correspondence:** Goran Gašparac (goran.gasparac@cirus.dhz.hr)

**Abstract.** The application of regional-scale air quality models is an important tool in air quality assessment and management. For this reason, the understanding of model abilities and performances is mandatory. The main objective of this research was to investigate the spatial and temporal variability of background particulate matter (PM) concentrations, to evaluate the regional air quality modelling performance in simulating PM concentrations during statically stable conditions and to investigate processes that contribute to regionally increased PM concentrations with a focus on Eastern and Central Europe. The temporal and spatial variability of observed PM was analysed at 310 rural background stations in Europe during 2011. Two different regional air quality modelling systems (offline coupled EMEP and online coupled Weather Research and Forecast-Chem) were applied to simulate the transport of pollutants and to further investigate the processes that contributed to increased concentrations during high pollution episodes. Background PM measurements from rural background stations, wind speed, surface pressure and ambient temperature data from 920 meteorological stations across Europe, classified according to the elevation, were used for the evaluation of individual model performance. Among the sea-level stations (up to 200 m), the best modelling performance, in terms of meteorology and chemistry, was found for both models. The underestimated modelled PM concentrations in some cases indicated the importance of accurate assessment of regional air pollution transport under statically stable atmospheric conditions and the necessity of further model improvements.

## 1 Introduction

The increased concentration of particulate matter (PM) in the ambient environment is associated with a significant impact on human health (Anderson, 2009; Heal et al., 2012; Peters et al., 2001; Pope et al., 2002; Samet et al., 2000; Samoli et al., 2005). Continuous exposure to PM is considered to be among the top 10 most significant risk factors for public health globally, including Europe (Prank et al., 2016). The elevated PM concentrations in the atmosphere have effects on the ecosystem (acidification, eutrophication) and visibility (e.g., Putaud et al., 2010). These also affect various meteorological processes such as cloud formation and radiation. Consequently, PM has been recognised as a strong climate forcer (e.g., Andreae et al., 2005;

Jiang et al., 2013) that also has an influence on Earth's heat balance through the direct radiative effects and cloud processes (Prank et al., 2016). European aerosol phenomenology studies (Van Dingenen et al., 2004; Putaud et al., 2004, 2010) have shown that the annual background of PM with aerodynamic diameter $\leq 2.5$ $\mu m$ ($PM_{2.5}$) and $\leq 10$ $\mu m$ ($PM_{10}$) concentrations for continental Europe are strongly affected by regional aerosol transport. For example, long-range transport has been attributed

to contributing up to about three-fourths of the total urban $PM_{2.5}$ concentrations in Finland (Karppinen et al., 2004; Pakkanen et al., 2001). A large fraction of the urban population is exposed to levels of $PM_{10}$ in excess of the limit values set for the protection of human health by national and international bodies. There have been numerous recent policy initiatives that aim to control PM concentrations to protect human health (EEA, 2015); yet high levels are reported regularly in different parts of the world (Kumar et al., 2015, 2016). The main problem in the assessment of $PM_{10}$ is in its diverse chemical composition across

Europe. Nitrate is a main contributor in Northwest (NW) Europe, mineral dust in south (S) Europe, desert dust from Africa over the Mediterranean, carbon in Central Europe and sea salt in coastal areas of Europe. The total residence time of PM in the atmosphere is highly dependent on precipitation, which influences the deposition processes. Conversely, wind speed plays an important role in both PM advection and the alteration of PM size and composition. $PM_{10}$ usually deposits at closer distances from its sources than smaller particles (e.g. Dimitriou and Kassomenos, 2014). On average, the residence time of fine particles

($PM_{2.5}$) is usually about 4-6 days as opposed to 1-2 days for coarser particles ($PM_{2.5-10}$). The typical distances for deposition from the sources are around 2000 to 3000 km for the fine particles and 500 to 1000 km for coarse particles (WHO, 2006). $PM_{10}$ can be emitted directly to the atmosphere from various natural and anthropogenic sources (primary $PM_{10}$) or can be produced through photochemical reactions in the atmosphere (secondary $PM_{10}$). In addition, wind-blown soil and re-suspended street dust contribute largely to the coarse particle fraction (Amato et al., 2009; Forsberg et al., 2005; Harrison and Jones, 2005;

Jeričević et al., 2012; Kumar and Goel, 2016; Luhana et al., 2004; Putaud et al., 2004). The contribution to PM emissions can be relevant at spatial scales ranging from local to regional including long-range transport (e.g., Juda-Rezler et al., 2011; Querol et al., 2004).

Air quality models (AQM) play a significant role in the assessment and management of air quality. These are widely used in public health cohort studies given that the measurements are expensive and usually represent limited and small areas, e.g., rural

areas, mountains (Ritter, 2013). Previous research on PM mass modelling (e.g., Vautard et al., 2007) identified the general underestimation of PM mass from large-scale models (grid spacing ~50 km) and the difficulties in capturing the observed seasonal variations in an urban location. The complexity of PM mass modelling was also introduced in Prank et al. (2016) where various modelling systems were compared – Unified European Monitoring and Evaluation Programme, EMEP (e.g., Simpson et al., 2012), LOTOS (e.g., Schaap et al., 2008), SILAM (e.g., Sofiev et al., 2008), CMAQ (Community Multi-Scale

Air Quality; Environmental Protection Agency) – which showed similar underestimations of PM concentrations. Applications of Weather Research and Forecast with its chemistry model, the WRF-Chem model (Grell et al., 2005), showed a relatively good comparison with measurements of the total PM mass over Europe (Tuccella et al., 2012) but the model did not capture the trends of PM compounds. Other studies (e.g., Saide et al., 2011) also indicated challenges in the modelling of PM mass, especially during statically stable atmospheric conditions, due to the choice of vertical and horizontal resolution as well as

the influence of vertical and horizontal diffusion coefficients during model setup (Jeričević et al., 2010). Furthermore, the

WRF-Chem model was extensively tested during the intensive evaluation of online coupled models of the second phase of the Air Quality Model Evaluation International Initiative (AQMEII, 2012). During the exercise, overall underestimation of PM concentrations for all the stations was found due to a relatively coarse grid spacing (23 km) together with the overestimation of wind speed, which can result in fast removal of pollutants from urban sources and underprediction of secondary organic aerosol

(SOA) and grid-scale precipitation (e.g., Baró et al., 2015; Forkel et al., 2015). The EMEP performance is evaluated through continuous yearly technical reports such as EMEP (2016). The most recent studies showed significant technical improvements with updated initial and boundary conditions as well as with newer model versions, which include various modifications in the chemistry modules. Throughout the performed extensive tests (Gauss et al., 2016), the model generally underestimated the observed annual mean $PM_{10}$ levels by 24%. However, there was an overall relatively good agreement (correlation coefficient,

$r = 0.74$) between modelled and measured annual mean $PM_{10}$ concentrations. Number of AQMs are currently available for practical applications. These models can be broadly divided into two main groups: offline and online models. The offline models consider solving separately meteorological conditions prior to chemistry during the simulation runs. There exists a huge variety of offline models such as the Comprehensive Air Quality Model with Extensions, CAMx (EVIRON, 2010), the Community Multi-scale Air Quality, CMAQ (U.S. Environmental Protection Agency), EMEP and LOTOS-EUROS (e.g.,

Solazzo et al., 2012). In contrast to offline models, the online models were developed to include the more consistent description of processes such as atmospheric turbulence and to use a more frequent update of the meteorological variables within the chemistry part of the model. There are other reasons for online coupling such as the ability to treat feedback processes between aerosols and airflows. Examples of online models include WRF-Chem, Environment: High-Resolution Limited Area Model (Enviro-HIRLAM), the Consortium for Small-scale Modelling Aerosols and Reactive Trace gases (COSMO-ART), and the

non-hydrostatic mesoscale atmospheric model with climate module (Meso-NH-C); e.g., Baklanov et al. (2014).

The main objective of this research was to investigate the spatial and temporal variability in background PM concentrations using one year of observed data, to evaluate the regional AQM performance in simulating PM concentrations during the colder part of the year and to analyse and evaluate the episode that occurred in November 2011 of regionally increased PM concentrations in Eastern and Central Europe (the Pannonian basin) during statically stable atmospheric conditions followed by drought

periods. In this particular case, the pollution problems appeared to be of considerable concern in Hungary, e.g., smog alerts were issued in Budapest and eastern Hungary, various cars with high environmental impact were banned from the roads, and a ban was also issued on items such as burning leaves and garden debris (https://thecontrarianhungarian.wordpress.com/2011/11/08/hungarian-news-digest-nov-7-2011/). Based on the analysis in Spinoni et al. (2015), the Pannonian basin was characterised as an area with increased drought frequency per decade during the period from 1950 to 2012. This can have a strong effect on

air quality problems, e.g., a dust-bowl effect (Stahl et al., 2016). Further assessment is conducted by applying two regional models: the offline Unified EMEP and the online coupled WRF-Chem in the simulation of PM mass transport. Model results are compared against observed concentrations at rural background sea-level, elevated, and mountain stations in Europe.

Throughout the analysis, the indication of problems is given in the application of both regional models in simulating PM concentrations at different elevations (sea-level, elevated and mountain stations). We provide an individual validation of widely

used different setups of the modelling systems without harmonisation of emission and meteorological input fields. This is a

different approach than in e.g., AQMEII exercises and enables an essential scientific baseline for choosing the appropriate model for future needs in terms of resolution, physical parameterisation, emission dataset and the complexity of orography representation in practical applications. Due to the complexity of air quality problems regarding PM, this work aims at filling the gaps in knowledge of regional modelling of PM over Eastern Europe in terms of less information about PM concentrations (EEA, 2013) and therefore low accuracy in the PM emission inventory and it fits in with addressed problems in most of the air quality plans in Europe (Miranda et al., 2015).

## 2 Methodology

### 2.1 Measurements

The measurements of $PM_{10}$ from the rural background stations were taken from two available air quality databases. These were *AirBase*, the European air quality database maintained by the European Environmental Agency https://www.eea.europa.eu/data-and-maps/data/airbase-the-european-air-quality-database-7, and the database developed under the EU-funded PHARE 2006 project *Establishment of Air Quality Monitoring and Management System*, where 12 new rural stations were established in Croatia for PM measurements in 2011. For this study, $PM_{10}$ concentrations were available from 6 rural background stations in Croatia. The monitoring stations were divided into three categories based on their elevation: (i) sea-level (altitude from 0 to 200 m), (ii) elevated (from 200 to 500 m), and (iii) mountain stations (>500 m) to examine the spatial variability of pollution and to test the model performance at different levels. The differentiation of stations with respect to their elevation is important when dealing with station representativeness in models. According to current knowledge, it is found that numerical models perform differently at higher altitudes. This is mostly related to the vertical resolution of the model within the boundary layer (Bernier and Bélair, 2012). With respect to the elevation, the total numbers of stations used for further analysis (Section 3.1) and model validation (Section 3.3) are shown in Table 1. When interpreting average observed yearly, seasonal and episode $PM_{10}$ concentrations, it is important to note that the majority of the surface stations are in Northern and Western Europe, while the elevated and mountain stations are situated in Central and Eastern Europe. The density of rural background stations varies geographically with a significantly greater number of stations in Western and Northern Europe compared to Central and Eastern Europe. The $PM_{10}$ measurements were acquired with different approaches: gravimetric method (EN12341) using high volume samplers (HVS) and low volume samplers (LVS), $\beta$-attenuation monitoring (e.g., Willeke and Baron, 1993), TEOMs (Tapered Element Oscillating Microbalances) measurements (e.g., Patashnick and Rupprecht, 1980) and by the optical particle counters of the GRIMM 180 instrument. The comparison of the $PM_{10}$ concentration data obtained by different measurement methods is still considered to be a complicated issue. The standard gravimetric method (EN12341) is a classic method of weighing the mass deposited on a filter. It is accepted as a standard reference method against which all other measurement methods are validated (Noble et al., 2001; EC, 2010). Although this is a standard method used for compliance reasons in the EU, there are numerous studies showing that chemical reactions between the air and the deposited particles, as well as within the aerosol mass, also compromise these measurements. The ambient temperature and relative humidity greatly influence the actual mass loaded on the filter (Allen et al., 1997; Eisner and Wiener, 2002; Pang et al., 2002). For example, aerosol particles can contain

up to 30% water at 50% relative humidity (Putaud et al., 2004). Conversely, calibration, temperature and humidity issues can create artefacts that must be taken into account for TEOMs and $\beta$-attenuation monitoring (Allen et al., 1997; Hauck et al., 2004). Lacey and Faulkner (2015) addressed three objectives for the treatment of uncertainties gained with PM measurements: estimate the uncertainty, identify the measurement with the greatest impact on uncertainty and finally determine the sensitiv-

5 ity of total uncertainty to all measured parameters. As is common in these types of studies, the authors did not consider the uncertainty of measurements in further analysis.

## 2.2 Statistical analysis

The evaluation of model performance is a comprehensive task. In order to evaluate and validate modelling performance, various statistical measures such as bias ($BIAS$), index of agreement ($IOA$), correlation coefficient ($r$), root mean square error

10 ($RMSE$),normalised mean square error ($NMSE$), systematic ($NMSE_{sys}$) and unsystematic ($NMSE_{unsys}$) normalised mean square error were used (Chang and Hanna, 2004):

$$BIAS = \left( \frac{\overline{M} - \overline{O}}{\overline{O}} \right) \times 100\% \tag{1}$$

$$BIAS = \overline{M} - \overline{O} \tag{2}$$

$$IOA = 1 - \frac{\sum_{i=1}^{N} (O_i - M_i)^2}{\sum_{i=1}^{N} \left( abs(M_i - \overline{O}) + abs(O_i - \overline{O}) \right)^2} \tag{3}$$

$$r = \frac{n \sum O_i M_i - \sum O_i \sum M_i}{\sqrt{n \sum O_i^2 - (\sum O_i)^2} \sqrt{n \sum M_i^2 - (\sum M_i)^2}} \tag{4}$$

$$RMSE = \sqrt{\frac{1}{n} \sum_{i=1}^{N} (M_i - O_i)^2} \tag{5}$$

$$NMSE = \frac{\overline{(O - M)^2}}{\overline{OM}} \tag{6}$$

$$NMSE_{sys} = \frac{4 \left( \frac{(\overline{O} - \overline{M})}{O.5(\overline{O} + \overline{M})} \right)^2}{4 - \left( \frac{(\overline{O} - \overline{M})}{O.5(\overline{O} + \overline{M})} \right)^2} \tag{7}$$

$$NMSE = NMSE_{sys} + NMSE_{unsys} \tag{8}$$

where *M* stands for model predictions and *O* for observations.

As there is no single best modelling performance measure, it is recommended by Chang and Hanna (2004) that a suite of different performance measures needs to be applied. Results should be carefully interpreted by taking into account advantages and disadvantages of all applied statistical measures and assuring that those are complementary to each other and leading to the same conclusion on the certain ability of the model performance. Therefore as already previously noted in this Section, a set of different statistical measures is used in order to understand the ability of the model to properly estimate high pollution episodes of $PM_{10}$ concentrations and to evaluate the relations between chemical and meteorological parameters. $BIAS$ refers to the arithmetic difference between *M* and *O* indicating model's general overestimation or underestimation of analysed parameters. It is known that a model whose predictions are completely out of phase with observations to still have a $BIAS = 0$ because of compensating errors. Different $BIAS$ was used: for evaluating model performance regarding $PM_{10}$ we used $BIAS$ under equation (1) as opposed to meteorological parameters under equation (2). $r$ and $IOA$ are dimensionless measure of model accuracy. $r$ is sensitive to a good agreement of extreme data pairs and a scatter plot might show generally poor agreement but the presence of a good agreement for a few extreme pairs will greatly improve $r$. The $IOA$ is the ratio of the mean square error and the potential error and then subtracted from one (Willmott, 1984). The $IOA$ varies from 0 to 1 with higher index values indicating that *M* have better agreement with the *O*. Although the $IOA$ provides some improvement over the $r$, it is still sensitive to extreme values due to the square differences in the mean square error in the numerator. $RMSE$ gives information on the spread of the residuals from the regression line, it highly depends on the magnitude of the parameter on which $RMSE$ is applied and therefore it cannot be compared with $RMSE$ of some other parameter. $NMSE_{sys}$ is a measure which with $NMSE_{unsys}$ provide information on systematic and unsystematic (random) errors in the model.

## 2.3 Boundary layer height determination

One of the widely used methods for deriving boundary layer height from numerical models is based on the assumption that turbulence collapses to laminar flow when the bulk Richardson number $Ri_B$, exceeds values of a critical $Ri_B$ (~0.25 and larger), and the height at which this occurs can be considered as a boundary layer height (Jeričević et al., 2010). Using sounding and modelled data, $Ri_B$ was calculated based on the following expression and shown in Section 3.3.3.

$$Ri_B = \frac{g\,(z - z_0)}{\overline{\theta\,(z)}} \frac{\theta\,(z) - \theta\,(z_0)}{(u\,(z))^2 + (v\,(z))^2} \tag{9}$$

where

$z$ is the height of the particular model level,

$z_0$ is the height of the first level in the model,

$\theta\,(z)$ is the potential temperature at the height $z$,

$\theta(z_0)$ is the potential temperature at the height $z_0$,

$\overline{\theta(z)}$ is the averaged potential temperature between the first level ($z_0$) and particular level ($z$)

$u(z), v(z)$ are the wind components on particular levels.

Comparison of estimated planetary boundary layer height (PBLH) was carried out using equation (9) rather than comparing the direct output of model-derived PBLH values as each model is using a different method for calculation of the PBLH. By using the same methodology for PBLH determination, uncertainties are reduced and the more realistic evaluation of two modelled PBLH values is assured.

## 2.4   Air quality models

This work is based on the intensive tests performed in Gašparac et al. (2016), where the WRF-Chem, Unified EMEP and WRF-CAMx models were evaluated against the surface measurement stations over Croatia under different atmospheric static stability conditions. Here, both EMEP and WRF-Chem AQMs are used to determine the spatial and temporal distribution of $PM_{10}$ concentrations, possible transboundary transport and to evaluate the performance of the individual model systems during a one-month period at the sea-level, elevated and mountain rural background stations in Europe.

### 2.4.1   EMEP model

The EMEP chemical transport model (Simpson et al., 2012), developed by the Meteorological Synthesizing Centre-West (MSC-W) was used to perform calculations of $PM_{10}$ concentrations (www.emep.int). The model domain encompassed all of Europe with a horizontal grid spacing of $50 \times 50$ km$^2$, extending vertically from surface level (first model level height around 42 m) to the tropopause at 100 hPa, as seen in the Supplementary Information (SI) Fig S1. The basic physical formulation of the EMEP model is derived from Berge and Jakobsen (1998). The model derives its horizontal and vertical grid from the input meteorological data. The daily meteorological input data used for the EMEP/MSC-W model for 2011 were based on experimental forecast runs with the Integrated Forecast System (IFS), a global operational forecasting model from the European Centre for Medium-Range Weather Forecasts (ECMWF). Vertically, the 60 eta levels of the IFS model were interpolated onto the 37 EMEP sigma levels. The emission input for the EMEP/MSC-W model, with a horizontal grid spacing of $50 \times 50$ km$^2$, consists of gridded annual national emissions based on emission data reported every year to EMEP/MSC-W (until 2005) and to the Centre on Emission Inventories and Projections (from 2006) by each participating country. The standard emissions input required by the EMEP model consists of gridded annual national emissions of sulphur dioxide ($SO_2$), nitrogen oxides ($NO_x =$ $NO + NO_2$), ammonia ($NH_3$), non-methane volatile organic compounds (NMVOC), carbon monoxide (CO), and particulates ($PM_{2.5}$, and $PM_{2.5-10}$). The PM categories can be further divided into elemental carbon, organic matter, and other compounds as required. Emissions can be set from anthropogenic sources such as the burning of fossil and biomass-based fuels, solvent release or from natural sources such as foliar VOC emissions or volcanoes. Several sources are challenging to categorise into anthropogenic versus natural categories (Winiwarter and Simpson, 1999), for example, emissions of NO from microbes in soils being promoted by N-deposition and fertiliser usage. The anthropogenic emissions are categorised into 11 SNAP (Selected Nomenclature for sources of Air Pollution) sectors based on their sources. Emission integration during simulation is

distributed vertically, based on the SNAP sectors and plume-rise calculations performed for different types of emission sources, and temporally, based upon time factors (i.e., monthly, daily, day-of-week, weekly, hourly).

Regarding the planetary boundary layer parameterisations under statically stable atmospheric conditions, EMEP includes a non-local vertical diffusion scheme based on a linear exponential profile with coefficients calculated from large eddy simulation

(LES) data and boundary layer height determined using the bulk Richardson number method (Jeričević and Večenaj, 2009; Jeričević et al., 2010; Simpson et al., 2012). Other mechanism used in this work (e.g. chemical scheme: EmChem09, chemical preprocessor: GenChem) are described in Simpson et al. (2012).

The above-written setup of EMEP model with the IFS meteorology as an initial and boundary meteorological conditions is later on referred and used in a form as "EMEP model". Any further comparison of meteorological conditions obtained in EMEP

simulations is related to the IFS model and $PM_{10}$ to the choice of EMEP chemistry parameterisation.

### 2.4.2    WRF-Chem model

The WRF-Chem model is the WRF (Weather Research and Forecasting) model (http://www.wrf-model.org) coupled with chemistry. It is a state-of-the-art air quality model (Grell et al., 2005) in which the chemistry (emission, transport, mixing, and chemical transformation of trace gases and aerosols) is simultaneously simulated with meteorology (online coupling). The

WRF is a mesoscale numerical weather prediction system designed for operational forecasting needs and atmospheric research (Skamarock et al., 2008). The model setup was based on earlier research (Gašparac et al., 2016; Grgurić et al., 2013; Jeričević et al., 2017) where the results were evaluated against measurements at meteorological stations in Croatia. In this paper, we used the WRF-Chem version 3.5.1. A Mercator projection was used in a one-domain run on 170 points in the east-west direction and 145 points in the north-south direction, with a cell size of $18 \times 18$ km$^2$ (*SI* Fig S1) and a vertical grid spacing encompassing

the atmosphere from surface level (first model level height around 22 m) to the height of ~23 km in 50 unequally sorted sigma levels that were more densely distributed near ground level. Initial and boundary meteorological conditions were provided by NCEP (National Centers for Environmental Prediction) Final Analysis (FNL ds083.2) with 1 degree of horizontal resolution and a time step of every 6 hours. They were selected based on previous research and other conducted studies with WRF or WRF-Chem model (Gašparac et al., 2016; Grgurić et al., 2013; Jeričević et al., 2017; Syrakov et al., 2015). FNL analyses are

a product of Global Data Assimilation System (GDAS) which continuously make multiple analyses of collected observational data from Global Telecommunications System (GTS) and various other sources. The whole analysis is available at 26 pressure levels from the surface to a height of ~28 km. The input emissions were prepared via the PREP-CHEM Sources tool (Freitas et al., 2011) with EDGAR (version 4.3.1., Emissions Database for Global Atmospheric Research) emission inventory for the year 2011. Biogenic emissions were calculated from MEGAN (Model of Emissions of Gases and Aerosols from Nature;

Guenther et al. (2006)) and lateral boundary and initial conditions were created from the global chemistry model MOZART (Emmons et al., 2010). The detailed WRF-Chem setup is shown in Table 2.

It is worth pointing out that the results of statistical analysis and model evaluation further on in the text will not describe the performance of the model itself, but rather will describe the performance of a set of selected parameterisations and chemical and meteorological initial and boundary conditions used in WRF-Chem model. Following this, when referring to the "WRF-

Chem model" in the text, the authors are referring to the WRF-Chem model with the above-described setup (Table 2). The WRF-Chem simulation is performed from 29 October to 30 November and EMEP from 1 October to 30 November. As all statistical analysis was done for dates after 1 November the simulation length was long enough to overcome the effects of spin up time.

## 3   Results

Available daily averaged rural background $PM_{10}$ concentrations ($\left(\overline{PM_{10}}\right)_d$) over Europe (Table 1) were analysed in the following sections with annual temporal variations and the episodes of very high $\left(\overline{PM_{10}}\right)_d$ concentrations that occurred during November 2011.

### 3.1   Analysis of PM measurements

We analysed the $\left(\overline{PM_{10}}\right)_d$ measurements from 310 stations over a period of one year during 2011. Following the air quality report in Europe (EEA, 2013), $\left(\overline{PM_{10}}\right)_d$ limit values (2008/50/EC Directive, LV=50 $\mu$g/m$^3$) were exceeded at both urban and rural sites in Europe during 2011. These "hotspots", locations with exceedances of the LV, were in South Poland, the Czech Republic, the Po Valley, the Balkan Peninsula, Portugal and Turkey. In this work, we focused on the area and rural background stations shown in Fig 1. The analysis of measurements from 310 rural background stations showed that observed $\left(\overline{PM_{10}}\right)_d$ exceeded LV 5456 times during 2011 and were mainly located in the hotspot areas (Fig 1). The seasonal variation in $\left(\overline{PM_{10}}\right)_d$ during 2011 was significant at the 5% level (based on analysis of variance, $ANOVA$; $p$=0). The applied $ANOVA$ is calculated via `scipy` python package. This particular one-way $ANOVA$ tests the null hypothesis that two or more groups have the same population mean. The $p$ value is common variable used in hypothesis testing, the smaller the $p$ value, the stronger is the evidence that hypothesis needs to be rejected (Heiman, 2001).

Spatially averaged seasonal values of $\left(\overline{PM_{10}}\right)_d$ were 21.62 $\mu$g/m$^3$, 21.74 $\mu$g/m$^3$, 14.96 $\mu$g/m$^3$ and 20.87 $\mu$g/m$^3$ for DJF, MAM, JJA and SON, respectively. Only during summer (JJA) a decrease was found with respect to other seasons over Europe. However, it should be noted that significant differences in PM levels across Europe are recognised (Putaud et al., 2004) and a deeper analysis of spatial and temporal variations in background $PM_{10}$ concentration is needed. Fig 2 presents individual $\left(\overline{PM_{10}}\right)_d$ values for each rural background station (lower panel), spatially averaged $\left(\overline{PM_{10}}\right)_d$ over the all stations (green line, upper panel) and the maximum $\left(\overline{PM_{10}}\right)_d$ values among all rural background stations (red line, upper panel) during 2011. The time series of these $\left(\overline{PM_{10}}\right)_d$ concentrations indicate the increase in concentrations at all rural background stations (Fig 2) during DJF and SON seasons (i.e., the colder part of the year). During these seasons, $\left(\overline{PM_{10}}\right)_d$ values at all rural background stations were relatively high, reaching 40 $\mu$g/m$^3$. During the colder part of the year, most of the stations recorded $\left(\overline{PM_{10}}\right)_d$ values above the permitted LV which is mainly due to increased emissions from domestic heating and industrial activities (EEA, 2013). Moreover, according to e.g., EEA (2013); Saarikoski et al. (2008) aside from the primary sources (natural and anthropogenic), the secondary inorganic aerosols (SIA) and secondary organic aerosols (SOA) vary substantially across Europe from season to season, which indicates the presence of various $PM_{10}$ sources. SIA contributions are mostly related to SON

– DJF, domestic heating and large combustion plants, while SOA contribution is rather related to MMA – JJA seasons, e.g., emissions from vegetation. This can explain the relatively high daily concentrations in the MMA season (Fig 2).

## 3.2 Analysis of PM measurements and meteorological conditions during episodes in November 2011

Further analysis of the observed and modelled $\left(\overline{PM_{10}}\right)_d$ values is focused on November 2011, as the highest $\left(\overline{PM_{10}}\right)_d$ concentrations were present during the colder part of the year and prevailing meteorological conditions enabled the accumulation of the pollutants in the lower layers of the atmosphere over Europe. According to Blunden et al. (2012), a strong high-pressure field was encompassing the area over Central and Southern Europe during November 2011. Moreover, this month was the coldest in 2011 and extremely dry; it was the driest month in Bulgaria and Serbia with less than 25% of the national total averaged precipitation. During the SON season in 2011, anticyclonic conditions prevailed and below-average precipitation conditions were recorded. Following Cindrić et al. (2016), the drought was present in the continental part of Croatia, encompassing the Pannonian basin and surrounding countries, and was characterised by extremely long duration. It started in February 2011 and reached the most intense extremely dry conditions in November, when an increase in $\left(\overline{PM_{10}}\right)_d$ was recorded at the majority of the analysed rural background stations (Fig 2). In Western Europe, the autumn season temperature was above average normal (1961-1990) and was characterized by prevailing high-pressure field. This was observed particularly in November during which monthly average temperature records were exceeded (e.g. UK, France and Switzerland reported their second warmest autumn in last 100 years). Contrary to the Western Europe, the increased nocturnal cooling decreased temperatures in Southeastern Europe. The dominating high-pressure field resulted in a decrease of precipitation in some Western and Central Europe countries, e.g. south France, Alpine region, Germany, Austria, Czech Republic, Slovakia, Hungary. All those countries reported the driest November in more than the last 100 years (Blunden et al., 2012). In order to identify the episodes and the areas of enhanced $\left(\overline{PM_{10}}\right)_d$ values, differences ($DF$) between the $\left(\overline{PM_{10}}\right)_d$ and annually-averaged $PM_{10}$ ($\left(\overline{PM_{10}}\right)_a$) at rural background stations were used, defined as:

$$DF = \frac{\left(\overline{PM_{10}}\right)_d - \left(\overline{PM_{10}}\right)_a}{\left(\overline{PM_{10}}\right)_a} \times 100\% \tag{10}$$

Spatial distribution of $DF$ values in percentage is shown in *SI* Fig S2. The significant increase in $\left(\overline{PM_{10}}\right)_d$ is defined as an increase in $DF$ of more than 100% with respect to the annual mean. If a significant increase in $DF$ was detected and lasted at least two consequent days, the area was identified as an area experiencing a high pollution episode. During November 2011, a significant increase in $\left(\overline{PM_{10}}\right)_d$ occurred generally over the addressed "hotspots" within the domain, and two high pollution episodes ($DF > 100\%$) were found, (Figs 3 – 4). During both episodes identified, the highest peaks (9 November in the first episode, Fig 3; 14 November in the second episode, Fig 4) occurred in the area of Central Europe and coastal part of Western Europe with $DF$ above 200%. Further on, observed meteorological conditions (daily averaged pressure field ($\left(\overline{mslp}\right)_d$), daily averaged surface temperature ($\left(\overline{t_{2m}}\right)_d$), daily averaged relative humidity ($\left(\overline{rh}\right)_d$), Figs 3 – 4; and daily averaged surface wind speed ($\left(\overline{ws}\right)_d$) and direction ($\left(\overline{wd}\right)_d$), Fig 5) along with $DF$ (Figs 3 – 4) were analysed to determine the mechanisms and relationships between the meteorology and the high pollution episodes.

At the beginning of November, values of $\left(\overline{PM_{10}}\right)_d$ were mainly at (or lower than) the mean monthly average values over most of the analysed stations while an increase in $DF$ ranging from 50 to 100% above annual averages was found over "hotspots" areas (South Poland, Czech Republic, Po Valley, Balkan Peninsula; *SI* Fig S2). On 3 November, cyclone Roft in Genoa bay generated intense rainfall in northern Italy (not shown). These conditions were followed by high S to SE winds over the Adriatic

Sea and nearby countries in the following days (Blunden et al., 2012). The characteristic meteorological conditions during or following Genoa low cyclones are strong flow aloft (Sirocco wind over the Adriatic Sea and Italy), rainfall in mid-Central Europe (Austria, Czech Republic and Poland) and the formation of high-$\left(\overline{mslp}\right)_d$ fields over Eastern Europe (Blunden et al., 2012). From 5 November, a first large-scale episode ($DF$>100%, *SI* Fig S2) started in Central and Northern Europe. The onset of the event was in Poland and Northeastern Germany and encompassed the coastal areas of Northern Europe, the Benelux

countries and Northern France in the following days until 9 November. During the first episode, a high-$\left(\overline{mslp}\right)_d$ field (Fig 3) formed over continental Europe, first affecting the east of Europe and gradually spreading to Western Europe. Over the affected area ($DF > 100\%$, Fig 3), the wind speed was generally reduced below 3 m/s, except at some isolated stations (Fig 5, left). Moderate to strong NE wind (5 – 6 m/s) started to blow in coastal and Northern Europe from 7 November until the end of the first episode when it turned to the ESE direction (Fig 5, left). Over the mountainous region in Central Europe (Czech

Republic, Slovakia and South Germany), the wind speed was persistent during the episode with relatively high magnitude (above 7 m/s) and generally in the SSE direction. Over the area with increased concentrations ($DF$>100%, Fig 3), a gradual moderate decrease in $\left(\overline{t_{2m}}\right)_d$ from east to west from the beginning to the end of the first episode was found (i.e., Poland < 0°C, Germany, the Czech Republic and Slovakia 0-5°C). On 10 November the wind speed was lower than 3 m/s over all of Europe (not shown), values were reduced and comparable to $\left(\overline{PM_{10}}\right)_a$ (*SI* Fig S2).

A building up of $\left(\overline{PM_{10}}\right)_d$ started again from 12 November (*SI* Fig S2 and Fig 4), mainly affecting stations in Central and coastal Western Europe. The observed concentrations exceeded the annual averages by up to 100%, ($DF$) affecting the areas with "hotspots" (Southern Poland, Czech Republic, Benelux countries) and up to 200% in Central Germany and Slovakia (Fig 4). In the following days, from 13 to 16 November, increased concentrations ($DF$> 100%) encompassing the area from Central Europe in the northwest direction through coastal areas in Germany, the UK and Ireland and were present in the southeastern

direction across the Czech Republic, Austria, Slovenia, Western Hungary and Croatia. During this, second episode, a high-$\left(\overline{mslp}\right)_d$ field again influenced the weather conditions (Fig 4). Low $\left(\overline{ws}\right)_d$ (<3 m/s; Fig 5, right) and a decrease in $\left(\overline{t_{2m}}\right)_d$ were found with the lowest $\left(\overline{t_{2m}}\right)_d$ measured in Eastern and Central Europe (Fig 4, below -5°C). Previously mentioned persistent conditions influenced the formation of statically stable atmospheric conditions during this episode (see section 3.3.3). Over particular areas with highly increased concentrations ($DF$> 200%, Poland, Germany, Slovakia, Czech Republic, Fig 4), an

increase in $\left(\overline{rh}\right)_d$ was found, except in the Pannonian basin (*SI* Fig S3) where relatively lower $\left(\overline{rh}\right)_d$ and higher $\left(\overline{t_{2m}}\right)_d$ values up to 20% and 5°C, respectively, were recorded in comparison with the surrounding areas. Moreover, within the areas of the Pannonian basin, a high $\left(\overline{mslp}\right)_d$and low wind speed conditions prevailed one day longer (Figs 4 – 5 right) in comparison with the surrounding areas. On 19 November a large-scale decrease in $\left(\overline{PM_{10}}\right)_d$ was detected and values of $\left(\overline{PM_{10}}\right)_d$ were reduced to those of $\left(\overline{PM_{10}}\right)_a$ at generally all stations (*SI* Figs S2, S6).

According to Figs. 3 – 4, during both episodes, mainly on all higher mountain stations within domain, the low-$\left(\overline{mslp}\right)_d$ was

observed. The $\left(\overline{mslp}\right)_d$ values were around 900 hPa which is common $\left(\overline{mslp}\right)_d$ for altitudes above 500 m. This means that in both cyclonic and anticyclonic conditions, the $\left(\overline{mslp}\right)_d$ was not disturbed and all processes such as advection, due to strong $\left(\overline{mslp}\right)_d$ gradients occurred mainly for sea-level and elevated stations.

## 3.3 Model evaluation

Numerical simulations using the EMEP (with a grid spacing of $50\times50$ km$^2$) and WRF-Chem (with a grid spacing of $18\times18$ km$^2$; *SI* Fig S1) models were provided for November 2011 to evaluate the performances of the individual, state-of-the-art models during November 2011 and to further investigate the processes contributing to the increased concentrations during the high pollution episodes. It is worth noting that differences between used emission databases were found in the spatial variability of PM$_{10}$ emissions and in the gridded input emission fields above the entire domains of EMEP and WRF-Chem.

Notable differences in emissions were found over the coastal areas and Eastern part of the domain particularly over Bosnia and Herzegovina, Serbia and Hungary which are crucial for the case studies analysed here. Aside from this, the difference in vertical resolution (first model level height – EMEP at 46 m, WRF-Chem at 22 m) can have a strong impact on surface concentrations and thus can be related to the differences in surface PM$_{10}$ concentrations obtained from the two used models.

### 3.3.1 Evaluation of model performances during November 2011

*Meteorological conditions*

    Vertical wind profile plays an important role in the dispersion of particulate matter. Hence, a validation of the modelled wind speed against measurements using mast-mounted instruments (Fig 6, Cabauw, Netherlands, $4.95^{\circ}$E, $51.97^{\circ}$N and Karlsruhe, in the western part of Germany, $8.39°$E, $48.98^{\circ}$N) was performed. During November there was no significant difference between modelled vertical profiles of wind speed below 75 m (Fig 6) for both sites. Modelled vertical wind profiles were close to mea-

surements at Cabauw site (up to 75 m), while at Karlsruhe the models underestimated the observed wind speed values in the first 180 m for WRF-Chem model and much higher above ground level for EMEP. The relatively coarse horizontal resolutions of the models have a great impact on wind values (e.g., Jeričević et al., 2012), which is why the modelled values correspond better to the observed wind values at the Cabauw site, situated in the flat terrain than to the values observed over the moderately complex terrain at the Karlsruhe site. Above 100 m, a change in the slope of the vertical wind speed profile for WRF-Chem

was found. The difference in model performance above the surface layer was previously addressed as to the proper choice of boundary layer parameterisation in Boadh et al. (2016).

    The modelled $\left(\overline{ws}\right)_d$, $\left(\overline{t_{2m}}\right)_d$, and $\left(\overline{mslp}\right)_d$ were compared to measurements from 920 synoptic stations within the domain taking into account the elevation of the station. A detailed statistical evaluation of the two individual model performances was conducted by calculation and analyses of six different statistical measures (Fig 7): $BIAS(\left(\overline{ws}\right)_d$, $\left(\overline{t_{2m}}\right)_d$, $\left(\overline{mslp}\right)_d$),

$IOA(\left(\overline{ws}\right)_d$, $\left(\overline{t_{2m}}\right)_d$, $\left(\overline{mslp}\right)_d$), $r(\left(\overline{ws}\right)_d$, $\left(\overline{t_{2m}}\right)_d$, $\left(\overline{mslp}\right)_d$), $RMSE(\left(\overline{ws}\right)_d$, $\left(\overline{t_{2m}}\right)_d$, $\left(\overline{mslp}\right)_d$), $NMSE_{sys}(\left(\overline{ws}\right)_d$, $\left(\overline{t_{2m}}\right)_d$, $\left(\overline{mslp}\right)_d$) and $NMSE_{unsys}(\left(\overline{ws}\right)_d$, $\left(\overline{t_{2m}}\right)_d$, $\left(\overline{mslp}\right)_d$). On the following Fig 7, individual scales for each analysed meteorological parameter are given as their magnitudes highly differs. Statistic measures calculated for wind speed are given in units m/s, temperature in $°$C and pressure in hPa. This is important for the interpretation of model scores in simulating different

meteorological parameters as e.g., $RMSE$ or $NMSE$ depend on their magnitude. Furthermore, the results from Fig 7 should be viewed as individual model performance rather than inter-comparison of two different model performances. According to $BIAS((\overline{ws})_d)$, the WRF-Chem model generally overestimated the observed $(\overline{ws})_d$, which is in accordance with other similar studies (e.g., Solazzo et al., 2012). The median of overestimation of $(\overline{ws})_d$ increases with the station altitude, $BIAS((\overline{ws})_d)$

was 1.8 m/s at sea level, 1.9 m/s at elevated and 2.8 m/s at mountain stations). WRF-Chem successfully predicted $(\overline{mslp})_d$ and $(\overline{t_{2m}})_d$ as $BIAS((\overline{mslp})_d, (\overline{t_{2m}})_d)$ values were very low at sea level and elevated stations while small to moderate $(BIAS((\overline{mslp})_d)\sim1.2$ hPa, $BIAS((\overline{t_{2m}})_d)\pm1°C)$ on mountain stations. The $BIAS((\overline{mslp})_d)$ increases with a height for both models. On elevated stations, a median of $BIAS((\overline{mslp})_d)$ decreased up to 1 hPa for both models, however for mountain stations it is in a range from -10 to 150 hPa for both models. Very low $(\overline{mslp})_d$ observed during particular high pollution

episode was not well represented in both models. EMEP model predicted $(\overline{ws})_d$ and $(\overline{mslp})_d$ well with low $BIAS$ values at sea-level and elevated station, while for surface $(\overline{t_{2m}})_d$ values, underestimation was found $(BIAS((\overline{t_{2m}})_d) \sim$ -2, 3, 4 °C at sea-level, elevated and mountain stations, respectively). The median $IOA((\overline{t_{2m}})_d)$ was relatively high for both models, while for $IOA((\overline{ws})_d)$ to small extent lower. For both parameters the decrease of performance with height was found. This indicates problems in simulations with regional models over complex terrain, which is confirmed by the values of $r$ that were consistent

for both models. As a result of small $BIAS((\overline{mslp})_d)$ over sea-level and elevated stations the $IOA((\overline{mslp})_d)$ was close to 1. However, over the mountain stations a high spread of values was found as the formulation of $IOA$ is very sensitive to the extreme values. The models did not show any substantial unsystematic errors for $(\overline{mslp})_d$. Systematic were as well low except on mountain stations. The range of both systematic and unsystematic errors increased with height for $(\overline{t_{2m}})_d$; the median values of $NMSE_{sys}$ and $NMSE_{unsys}((\overline{t_{2m}})_d)$ for the EMEP model were the highest for elevated stations. In the case of the

WRF-Chem model, $NMSE_{sys}((\overline{t_{2m}})_d)$ increases with height, while for the EMEP model, the highest $NMSE_{unsys}((\overline{t_{2m}})_d)$ median was found at elevated stations.

Overall, during a one-month period of simulation, EMEP had the lowest systematic errors for $(\overline{ws})_d$, while WRF-Chem had the lowest systematic errors for $(\overline{t_{2m}})_d$. Based on given statistic, overall model performance regarding meteorological parameters was in accordance to similar modeling studies. For example, negative $BIAS$ and high $r$ for $(\overline{t_{2m}})_d$ was found in (e.g., Skjøth

et al., 2015; Qu et al., 2014). Positive $BIAS$ for $(\overline{ws})_d$ was already addressed as an issue in related studies such as e.g., Baró et al. (2015); Forkel et al. (2015), while results for $(\overline{mslp})_d$ for sea-level and/or elevated stations are in accordance with e.g., Qu et al. (2014).

*Chemistry*

The modelled $(\overline{PM_{10}})_d$ values were compared with the available corresponding measurements (Table 1) with respect to

height by applying statistical measures (Fig 8, *SI* Table S1). Although the number of stations varies within altitude groups (Table 1), the overall model performance can be inferred from the addressed figure and table. The underestimation of concentrations was found at sea-level (the median of -44% and -26% for the WRF-Chem and EMEP models, respectively) and elevated stations (-55% and -29% for the WRF-Chem and EMEP models, respectively; *SI* Figs S4-S5). At mountain stations, EMEP had good agreement of ~13%, while underestimation with respect to WRF-Chem is still present ~ 33%. According to *SI* Figs S4 – S5,

the $BIAS((\overline{PM_{10}})_d)$ in both model simulations showed a similar distribution with respect to the height of the station, i.e.,

moving from underestimation towards overestimation. $IOA((\overline{PM_{10}})_d)$ was generally equally persistent with height for both models (Fig 8) with a somewhat higher score for simulations with the EMEP model except for the sea-level stations where the median of both models had equal value (0.9, *SI* Table S1). The highest $r((\overline{PM_{10}})_d)$ values were above 0.87 for both models; however, the overall performance in terms of $r$ (median, *SI* Table S1) for both models was relatively low, particularly for the

elevated and mountain stations. The average values over the domain for the WRF-Chem and EMEP models were 0.39, 0.21 and 0.19, and 0.48, 0.28 and 0.24 for sea-level, elevated and mountain stations, respectively. High variability in $r$ values over the domain for both models is found (*SI* Figs S4 – S5). As $r$ is a measure of linearity and is highly dependent on the estimation of peak values and trends, the low values at all stations are attributed to mismatch of modelled and measured peak values during the period of analysis. Even a small discrepancy between measured and modelled $(\overline{PM_{10}})_d$ can lead to a decrease in $r$.

$RMSE(\overline{PM_{10}})_d)$ decreases with height, and the highest median $RMSE$ values were found over sea-level stations (20.7 for WRF-Chem model, and 17.3 for the EMEP model; Fig 8, *SI* Table S1). It should be noted that $RMSE$ highly depend on the concentration magnitudes. Higher values of $RMSE$ for both models generally correspond to the stations with low $r$ values (the hotspot areas: Southern Poland, Czech Republic, Po valley; *SI* Figs S4 – S5. Fig 8 shows that the trends of systematic errors differ between the models. The lowest errors in the WRF-Chem model were found over sea-level stations, while the highest

over elevated stations. The errors in the EMEP model were comparable at all altitudes; however, the range of errors increased with height. Similar performance was found for the EMEP model and unsystematic errors. The median values were comparable at all altitudes, while the range slightly increased with height. In the case of the WRF-Chem model, a moderate increase in the median and the range of unsystematic errors with height was found. The areas affected with increased $NMSE_{unsys}((\overline{PM_{10}})_d)$ were the hotspot areas (Po Valley and Southern Poland) in the EMEP model, while in the WRF-Chem model, the increase in

$NMSE_{unsys}((\overline{PM_{10}})_d)$ is found at almost all stations, particularly at the mountain level (*SI* Figs S4 – S5). It must be pointed out that both $NMSE_{sys}$ and $NMSE_{unsys}$ of $(\overline{PM_{10}})_d$ in the EMEP model were substantially smaller at all altitudes with respect to the WRF-Chem model.

The overall performance of the models regarding $(\overline{PM_{10}})_d$ was good, and the results are in agreement with similar modelling studies (e.g. Werner et al., 2015; Baró et al., 2015; Forkel et al., 2015; Gauss et al., 2016). Due to the coarser grid resolutions,

differences in terrain height could lead to a problem in station representativeness in regional models. Generally, from the given analysis, it can be concluded that the performance of both models varies with height. There is a moderate agreement in all of the analysed meteorological parameters and $(\overline{PM_{10}})_d$, which shows a trend in the decrease in performance with a height. This can be seen in Figs 7 - 8. The better modelling performance was found for $(\overline{t_{2m}})_d$ using the WRF-Chem model, while for $(\overline{ws})_d$ in the case with the EMEP model. Both systematic and unsystematic errors for $(\overline{PM_{10}})_d$ were the lowest for

sea-level stations and at comparable levels between models. Values of $r((\overline{PM_{10}})_d)$ and $RMSE((\overline{PM_{10}})_d)$ decreased with height for both models. A substantial number of elevated stations are located in the vicinity of hotspot areas (south Poland, Czech Republic, etc.; *SI* Figs S4, S5) and are therefore subject to a strong influence from high emissions. This can explain the relatively lower performance (e.g., $NMSE_{sys}((\overline{PM_{10}})_d)$ for the WRF-Chem model; $RMSE((\overline{PM_{10}})_d)$ for both applied models) of a number of stations at an elevated level with respect to other altitudes in this area.

### 3.3.2 Analysis of model performance during the large-scale episodes

Here we focus on the analysis of spatial and temporal variations in the mean surface daily fields $(\left(\overline{mslp}\right)_d, \left(\overline{t_{2m}}\right)_d, \left(\overline{pblh}\right)_d,$ $\left(\overline{ws}\right)_d$ with $\left(\overline{wd}\right)_d)$ between the two applied models in order to investigate the mechanisms behind the high pollution episodes. In Fig 9, the modelled surface $\left(\overline{PM_{10}}\right)_d$ together with $\left(\overline{mslp}\right)_d, \left(\overline{t_{2m}}\right)_d, \left(\overline{pblh}\right)_d, \left(\overline{ws}\right)_d$ with $\left(\overline{wd}\right)_d$ for the two days with peak $\left(\overline{PM_{10}}\right)_d$ concentrations (9 and 14 November 2011) during the two high pollution episodes obtained with the EMEP and WRF-Chem models are shown. The distribution of $\left(\overline{t_{2m}}\right)_d$ for both selected days was generally equal over the entire domain for both models. The $\left(\overline{pblh}\right)_d$ tends to have lower values (<100 m) in the WRF-Chem simulation and gradients in the pressure fields are much higher in comparison with the EMEP model. Values of $\left(\overline{ws}\right)_d$ were generally higher within the domain for the WRF-Chem simulation. However, both models indicated the same areas with lowered wind speed, which is in accordance with the measurements (Fig 5). Generally, both models correctly indicated areas affected by high pollution episodes ($DF$>100%, Figs 3 – 4). Over areas with $\left(\overline{pblh}\right)_d$ below 100 m, peaks of $\left(\overline{PM_{10}}\right)_d$ were found, reaching measured $\left(\overline{PM_{10}}\right)_d$ values (*SI* Fig S6). For both peak days the models are consistent, showing prevailing high $\left(\overline{mslp}\right)_d$ fields, relatively cold areas with low $\left(\overline{pblh}\right)_d$ (more evident in the case of the WRF-Chem model) and low $\left(\overline{ws}\right)_d$ conditions (more evident in the EMEP model) over the affected areas with $\left(\overline{PM_{10}}\right)_d$ concentrations (Figs 3 – 4). The *SI* Tables S1 – S2 are showing the minimum, maximum and median values of $\left(\overline{PM_{10}}\right)_d, \left(\overline{t_{2m}}\right)_d, \left(\overline{pblh}\right)_d, \left(\overline{mslp}\right)_d, \left(\overline{ws}\right)_d$ over the domain (Fig 1) for both models during episodes. Minimum, maximum and median values of $\left(\overline{mslp}\right)_d$ between models were similar. Average minimum $\left(\overline{mslp}\right)_d$ over domain was 1004.77 hPa and 1005.55 hPa, average maximum 1031.93 hPa and 1031.44 hPa and average median 1021.18 hPa and 1020.33 hPa for WRF-Chem and EMEP model respectively. The average minimum $\left(\overline{t_{2m}}\right)_d$ for WRF-Chem ~-5.54°C was lower in respect to EMEP model ~-2.31°C, however average maximum $\left(\overline{t_{2m}}\right)_d$ ~20°C and median $\left(\overline{t_{2m}}\right)_d$ ~10°C values were same for both models. $\left(\overline{pblh}\right)_d$ in WRF-Chem model varied from an average minimum value of 38.97 m to an average maximum value of 1612.29 m, while EMEP had much higher average minimum value 137.62 m (due to coarser vertical resolution of the EMEP model) and somewhat lower average maximum value ~ 1585.81 m (*SI* Tables S1 – S2). $\left(\overline{ws}\right)_d$ was more variable over the domain for WRF-Chem in respect to the EMEP model. During both episodes, minimum $\left(\overline{ws}\right)_d$ in WRF-Chem was in the range from 0 to 0.11 m/s, while maximum varied from 19.77 m/s up to 36.34 m/s, the average median $\left(\overline{ws}\right)_d$ was 5.00 m/s. For EMEP model, minimum $\left(\overline{ws}\right)_d$ was similar to WRF-Chem, and in the range from 0.01 m/s to 0.18 m/s, while maximum $\left(\overline{ws}\right)_d$ was lower than obtained with WRF-Chem simulation, in the range from 12.74 m/s to 16.77 m/s. Same was as well as for the average median $\left(\overline{ws}\right)_d$ lower than in WRF-Chem simulation, 3.60 m/s. The average $\left(\overline{PM_{10}}\right)_d$ concentrations were generally higher in the EMEP model. The average minimum $\left(\overline{PM_{10}}\right)_d$ concentrations were between 0.19 and 1.51 $\mu$g/m$^3$, average maximum $\left(\overline{PM_{10}}\right)_d$ was 62.04 $\mu$g/m$^3$ and 84.45 $\mu$g/m$^3$ and average median $\left(\overline{PM_{10}}\right)_d$ values were between 6.91 $\mu$g/m$^3$ and 13.46 $\mu$g/m$^3$ for WRF-Chem and EMEP model respectively during both episodes. The absolute maximum concentration obtained with the WRF-Chem model was 63.55 $\mu$g/m$^3$and 81.32 $\mu$g/m$^3$ while for the EMEP model, 110.09 $\mu$g/m$^3$ and 97.84 $\mu$g/m$^3$ during the first and second episode, respectively.

During the first episode, the presence of cyclone Ruft in the Gulf of Genoa was evident in both models (*SI* Figs. S7 – S8). Stronger surface winds occurred in the WRF-Chem simulation over Europe compared to the EMEP simulation, which con-

sequently resulted in different dynamics within the boundary layer (*SI* Fig S9). The onset of the high pollution event was in Central Europe in the EMEP model as shown in the measurements, but with lower concentrations with respect to the measurements (Fig 3 and *SI* Fig S7). With NE winds over the coastal areas of Northern Europe, the pollution was gradually spread to Western Europe. In the WRF-Chem model, the higher surface wind speed over Central Europe was well estimated (Fig 5; Fig *SI* Fig S7) and surface wind speeds over coastal areas in Northern Europe were well-represented in the second part of the episode, leading to a good estimation of potential transport of $\left(\overline{PM_{10}}\right)_d$ to Western Europe (*SI* Fig S8). This agrees with similar studies where the dependence of $\left(\overline{PM_{10}}\right)_d$ on $BIAS((\overline{ws})_d)$ was identified (e.g. Solazzo et al., 2012). During both episodes, the $\left(\overline{mslp}\right)_d$ on the synoptic-scale was correctly predicted by both models over the domain (*SI* Figs S6 – S7). Aside from the $(\overline{ws})_d$, notable differences between models performances were found in $\left(\overline{pblh}\right)_d$ (up to 200m) and $\left(\overline{t_{2m}}\right)_d$ (up to 5°C), which had an impact on the distribution and magnitude of the estimated high $\left(\overline{PM_{10}}\right)_d$ concentrations in both episodes. In simulations with the WRF-Chem model (*SI* Fig S11), the onset of the second episode was delayed up to 1.5 day in comparison with the measurements (Fig 4). Moreover, in the second episode, over areas with increased concentrations in Central Europe, the decrease of $\left(\overline{pblh}\right)_d$ followed by weak wind speed was found in accordance with the measurements (Fig 5, right, *SI* Fig S12). Recognised statically stable conditions (elaborated in Section 3.3.3.) with the presence of colder days prevailed over all of Europe. This favoured the build-up of concentrations in Northwest and Central Europe affecting all of Central Europe (*SI* Figs S8 – S9). The representation of meteorological conditions over the affected areas ($DF$>100%, Figs 3 – 4) agreed well with measurements during both episodes (Figs 3 – 5, *SI* Figs S7 – S12). Although differences in $(\overline{ws})_d$ were found between the models (*SI* Figs S9-S12), the areas with increased $\left(\overline{PM_{10}}\right)_d$ were appropriately similar. However, as previously pointed out, the models underestimated the measured surface concentrations (*SI* Figs S6 – S8).

### 3.3.3   Intercomparison of modelled PBL height against radio soundings

More detailed analyses of model results and the influence of meteorological parameters during the second episode were made against measurements within the area of the Pannonian basin (*SI* Fig S3). Pannonian basin endured high pollution events during the second high pollution episode that were mainly found at urban stations (not shown) due to the lack of rural background measurements. In the analysed period, increased values of $\left(\overline{PM_{10}}\right)_d$ can be depicted only on one available rural background station in the area, Fig 4. The increased concentrations can be observed also from modelling results (Fig 9, *SI* Fig S2). The area of increased concentrations is in accordance with the area of weak wind conditions (Fig 5) and low $\left(\overline{pblh}\right)_d$ values and can be described as an area of potentially statically stable conditions. The mean modelled vertical profiles during episodes at all available sounding stations within the area of interest agreed well with the measurements, except at the Belgrade station where both models underestimated wind speed by up to 10 m/s in first 2000 m (*SI* Fig S13).

Using sounding measurements and equation 9, the $Ri_B$ and boundary layer height ($H_{bl}$) were calculated for four sites within the Pannonian basin (*SI* Fig S3) and are shown in Fig 10. The same parameters were calculated from the WRF-Chem and EMEP modelling data. It must be pointed out that available sounding measurements were instantaneous values at 00 UTC only, while time step in WRF-Chem model was 1 hour and in EMEP 3h. The $Ri_B$ values calculated from soundings and modeled data shown on Fig 10 are represented with the same time step as input data: 12h for measurements, 1 h for WRF-Chem and 3

h for EMEP model. According to Fig 10, the models were consistent in $Ri_B$ and in estimating $H_{bl}$. The development of the atmospheric boundary layer started early in the morning with sunrise and reached values up to 350 – 400 m around 14:00 (local time), except between 17 and 21 November when a decrease in $H_{bl}$ was found. During this period the peak values of $H_{bl}$ reached 200 m and the statically stable conditions ($Ri_B$ >0.25) were dominant (light blue to dark blue color up to value of 2, above in white colour). As a strong increase in statically stable conditions occurred at all four stations, which are spread out within the basin, it can be concluded that statically stable conditions prevailed over the Pannonian basin during this particular event. A similar conclusion comes from values of $Ri_B$ and $H_{bl}$ calculated from soundings (Fig 10). Due to the coarse vertical resolution in some periods and low time step (24 hours), the contours are rough and the effect of sunrise on the development of $H_{bl}$ cannot be seen. However, at all four measurement stations prevailing statically stable conditions during the second-high pollution episode were indicated, which is in accordance with the modelling results.

## 4   Summary and conclusions

Numerical modelling of $\left(\overline{PM_{10}}\right)_d$ with different AQMs is still challenging (Baró et al., 2015; Prank et al., 2016; Laurent et al., 2016). It is therefore important to further analyse the different performances of regional models that have been widely used in practical applications. The main task of the current work was to investigate one of the weakest model capabilities, i.e., the simulations of AQMs under statically stable boundary conditions (e.g. Gašparac et al., 2016; Grisogono and Belušić, 2008) focusing on dynamic model aspects during episodes of elevated $\left(\overline{PM_{10}}\right)_d$ concentrations over Central and Eastern Europe. Here, two different regional AQMs, namely, EMEP and WRF-Chem, were applied to evaluate their individual state-of-the-art performance and to investigate the processes that contributed to a high $\left(\overline{PM_{10}}\right)_d$ concentration during pollution episodes that occurred in Europe. Other model intercomparison research studies over Europe and North America were done within the AQMEII project (e.g. Im et al., 2015; Solazzo et al., 2012; Rao et al., 2011). However, with respect to those large exercises with harmonised input data (same meteorology, emissions, boundary and initial conditions), the focus of this research was on the specific meteorological situations when statically stable atmospheric conditions prevailed accompanied by the occurrence of high $\left(\overline{PM_{10}}\right)_d$ concentrations. The offline EMEP and online WRF-Chem modelling systems were used with the available input data that are usually implemented in practical applications (e.g., environmental assessment studies). The added value here is in the individual statistical evaluation of such modelling systems using data from the large number of meteorological and air quality stations in Eastern Europe that have been less represented in other similar exercises. The analysed and modelled meteorological parameters were validated using surface measurements from 920 synoptic stations, soundings within the Pannonian region and mast-mounted instrument measurements. The $\left(\overline{PM_{10}}\right)_d$ concentrations were validated against surface measurements from 310 rural background stations. During the colder part of the year, when usually higher PM concentrations are observed, following model features are established:

- According to the low systematic errors a very good model performance is found in simulating $\left(\overline{mslp}\right)_d$ over sea-level and elevated stations, while high positive BIAS for both models was obtained over mountain stations.

- Good performance in modelling $\left(\overline{ws}\right)_d$ in EMEP and $\left(\overline{t_{2m}}\right)_d$ in the WRF-Chem model is found while on contrary EMEP model highly overestimated $\left(\overline{t_{2m}}\right)_d$, and WRF-Chem overestimated $\left(\overline{ws}\right)_d$.

- The differences in boundary layer dynamics were found in models through the analysis of vertical wind profiles.

- Based on calculated values of $Ri_B$, the evaluation of modelled $\left(\overline{pblh}\right)_d$ agreed well with the measurements for both models. However, according to the spatial $\left(\overline{pblh}\right)_d$ fields, derived directly from model, the WRF-Chem model generally tends to estimate lower $\left(\overline{pblh}\right)_d$ with respect to the EMEP model over areas affected by high pollution ($DF > 100\%$).

- From the results of the simulation of a one-month period that encompassed various meteorological conditions and different terrain types, we found:

  o Strong influence of meteorological conditions on increased background $\left(\overline{PM_{10}}\right)_d$ and correct estimation of the $\left(\overline{ws}\right)_d$ is recognised as one of the main factors in the dispersion of $\left(\overline{PM_{10}}\right)_d$.

  o General underestimation of background $\left(\overline{PM_{10}}\right)_d$ concentrations with both models, except with EMEP for mountain stations (slight overestimation).

  o Statistical analysis with respect to the terrain type shows the best modelling performance of $\left(\overline{PM_{10}}\right)_d$ and meteorology over sea-level stations (flat terrain). Both models tend to agree in decrease in performance with height, indicating problems in regional model simulations over complex terrain.

- From the analysis of the high pollution episodes, we can conclude following:

  o During the first high pollution episode, a high $\left(\overline{ws}\right)_d$ in the WRF-Chem model resulted in a decrease in surface $\left(\overline{PM_{10}}\right)_d$ while favourable conditions prevailed for the build-up of concentration in Central Europe over hotspot areas with a decrease in surface $\left(\overline{ws}\right)_d$.

  o Low wind speed conditions during the entire second episode, followed by high $\left(\overline{mslp}\right)_d$ and low $\left(\overline{pblh}\right)_d$, prevailed over the affected area ($DF > 100\%$).

  o Statically stable conditions were recognised as the main mechanism for the build-up of concentrations during the second episode. Both models produced low values of $\left(\overline{pblh}\right)_d$, (<100m in WRF-Chem and 100 – 200m in EMEP) over areas where stations recorded $\left(\overline{PM_{10}}\right)_d$ concentrations > 200% ($DF$) with respect to the annual mean (Figs 3 – 4, *SI* Figs S8, S12).

  o Underestimation of background $\left(\overline{PM_{10}}\right)_d$ concentrations with regional models is in accordance with other modelling studies (Gauss et al., 2016; Forkel et al., 2015).

- Reasons for the underestimation of modelled $\left(\overline{PM_{10}}\right)_d$ concentrations were attributed to the uncertainty of associated and inadequate treatments of formation processes that usually omit some components of atmospheric aerosols (e.g., SOA, SIA) and thus fail to estimate the total PM budget properly.

Dynamic model properties are very important: horizontal and vertical model resolutions and the boundary layer parameterisations in statically stable atmospheric conditions should be selected carefully. Furthermore, model simulations using more accurate emission inventory and larger (nested) domains with the finer resolution are necessary for further improving the model predictions. Future work using longer periods of simulations for both models, including other pollutants (e.g., NOx, SOx, PM

5    compounds and $O_3$) is recommended to make comparisons under various meteorological conditions.

*Code and data availability.* *AirBase*, the European air quality database is mantaned and available at http://acm.eionet.europa.eu/databases/airbase. WRF-Chem model is available at https://ruc.noaa.gov/wrf/wrf-chem/ and EMEP model at https://www.emep.int/index_model.html.

*Author contributions.* GG wrote the paper with the contribution from all authors. GG made simulations with the WRF-Chem model and made analysis of modeling and measurement data including preparation of all figures. JA made simulation with the EMEP model and

10   contributed in modeling analysis and preparation of figures and text. PK and BG contributed in data analysis and interpretation of the results.

*Competing interests.* The authors declare that they have no conflict of interest.

*Acknowledgements.* This work has been supported by the Croatian Science Foundation under contract IP-2013-11- 5928 (project ADAM-ADRIA). We would like to express our gratitude to Josip Križan and Ana Jurjević for comments on script programming and statistical evaluation that greatly improved the first stages of our work. We thank Gekom Ltd. for technical support and dr.sc. Svetlana Tsyro on devices

15   regarding EMEP model.

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

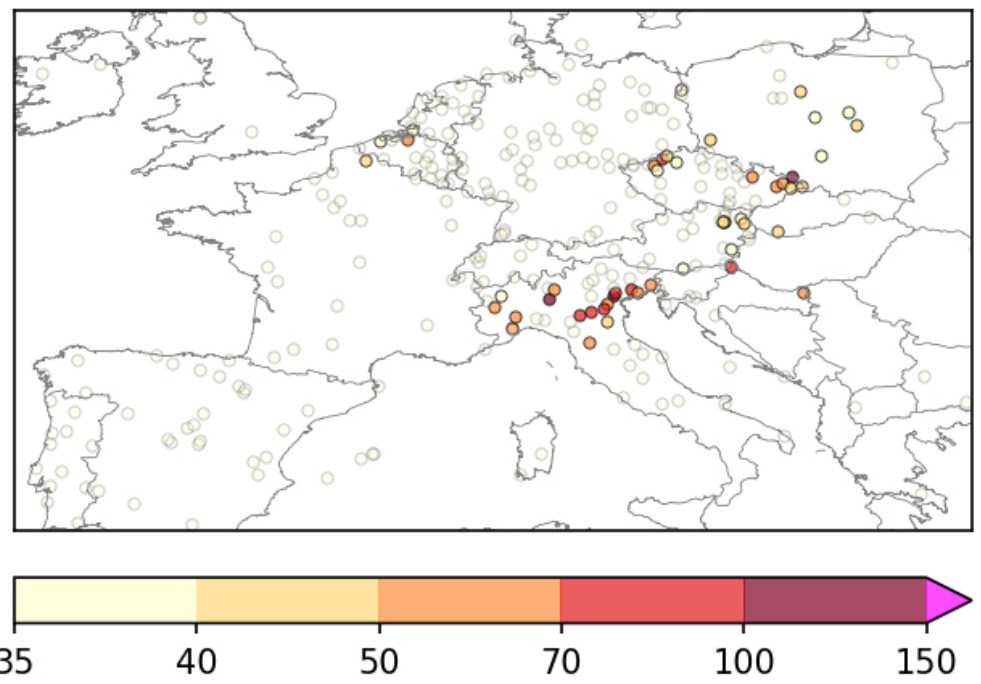

**Figure 1.** Number of days exceeding the daily limit PM$_{10}$ value (LV) at rural background stations during the year 2011 in the domain of the research. Stations marked with a grey circle represent less than or equal to 35 permitted exceedances during the year (2008/50/EC Directive).

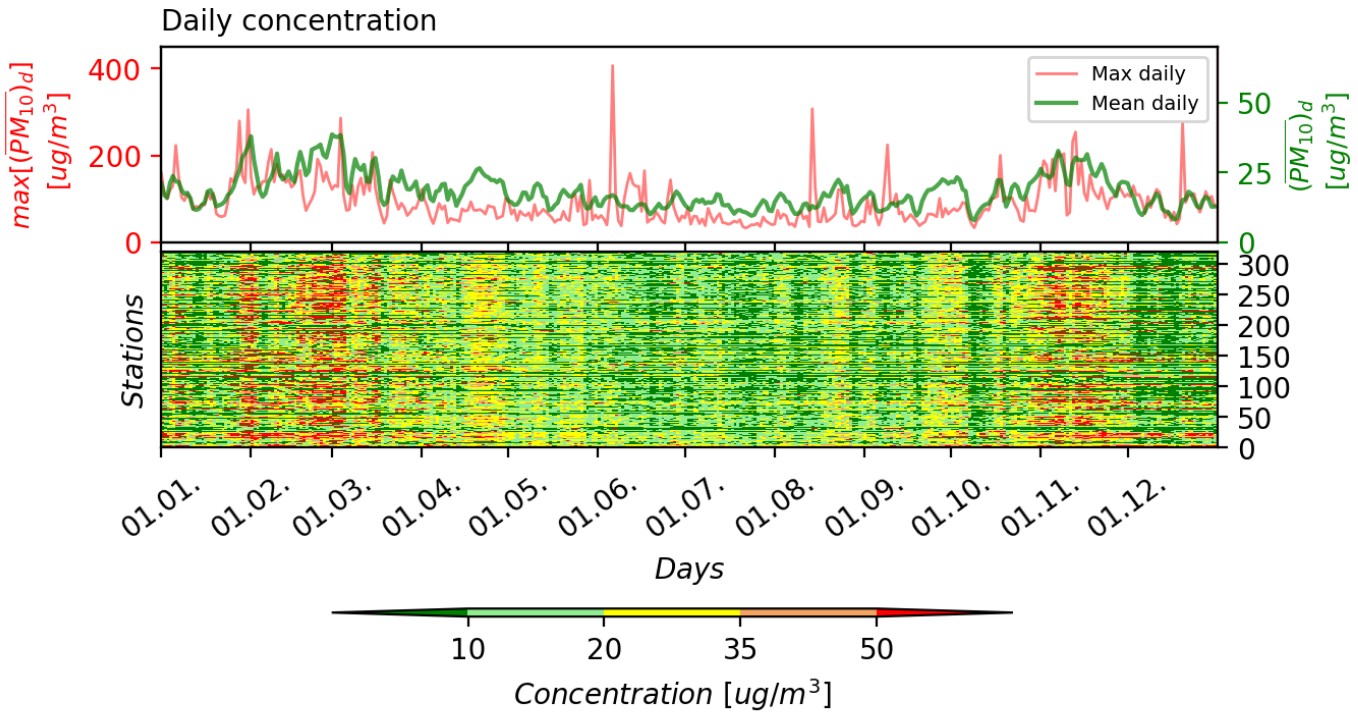

**Figure 2.** The spatially average (upper panel) over all the rural background stations (the green line, corresponding to the right green y-axis) and the maximum of $\left(\overline{PM_{10}}\right)_d$ for all rural background stations (the red line, corresponding to the left red y-axis) and $\left(\overline{PM_{10}}\right)_d$ (lower panel) during 2011. The values above 50 $\mu$g/m$^3$ (red colour) represent values above the daily limit values for PM$_{10}$ under the 2008/50/EC Directive.

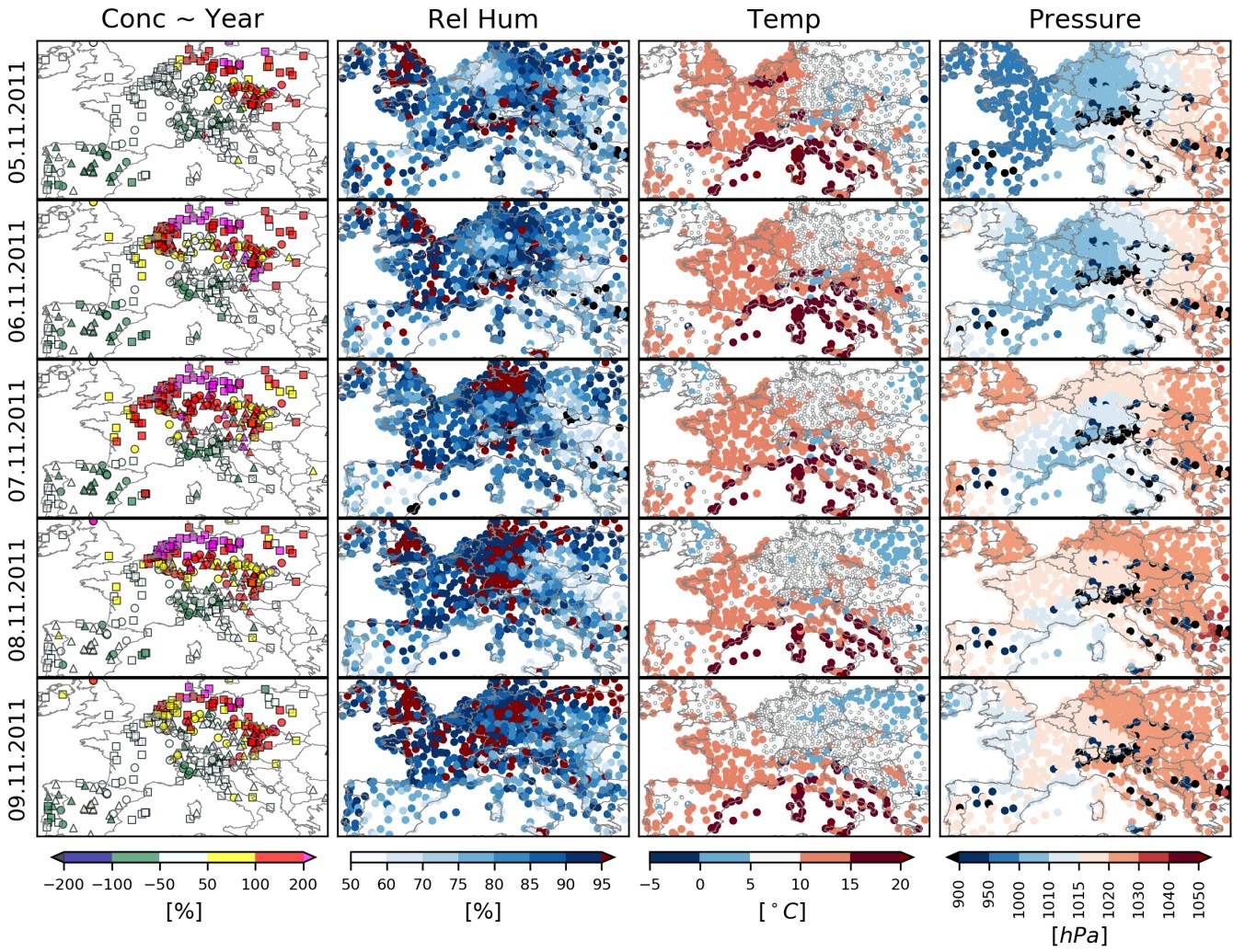

**Figure 3.** $DF$ and measurements from synoptic stations (relative humidity (Rel Hum), ambient temperature at 2m (Temp), and surface pressure) from the National Center for Environmental Prediction, Final Analysis (ds083.2) data during the first large-scale episode (5 to 9 November). Stations with a temperature between $0 - 5°C$ are marked with little grey dots due to better representativeness on the map.

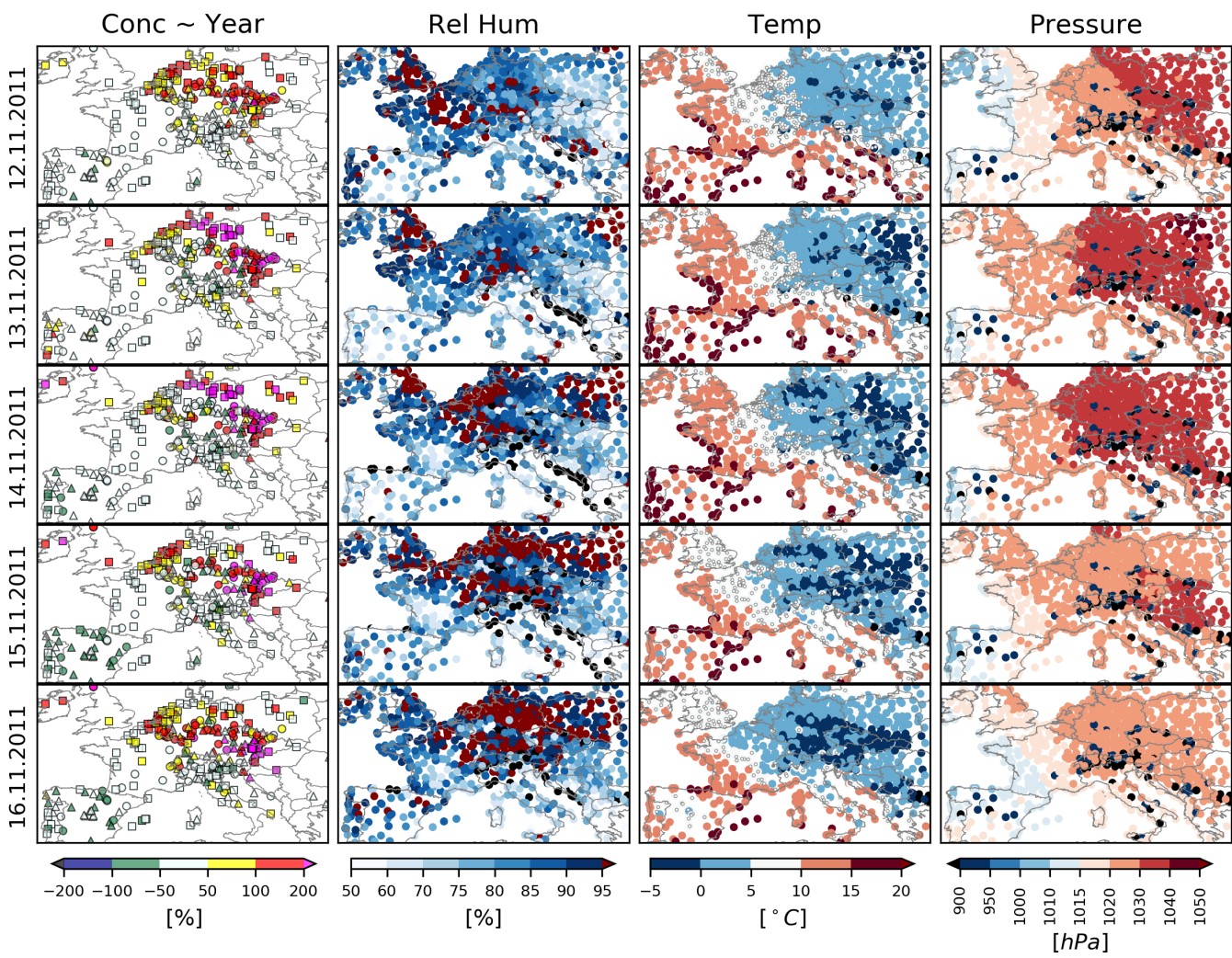

**Figure 4.** Same as Fig 4, but during the second large-scale episode (12 to 16 November).

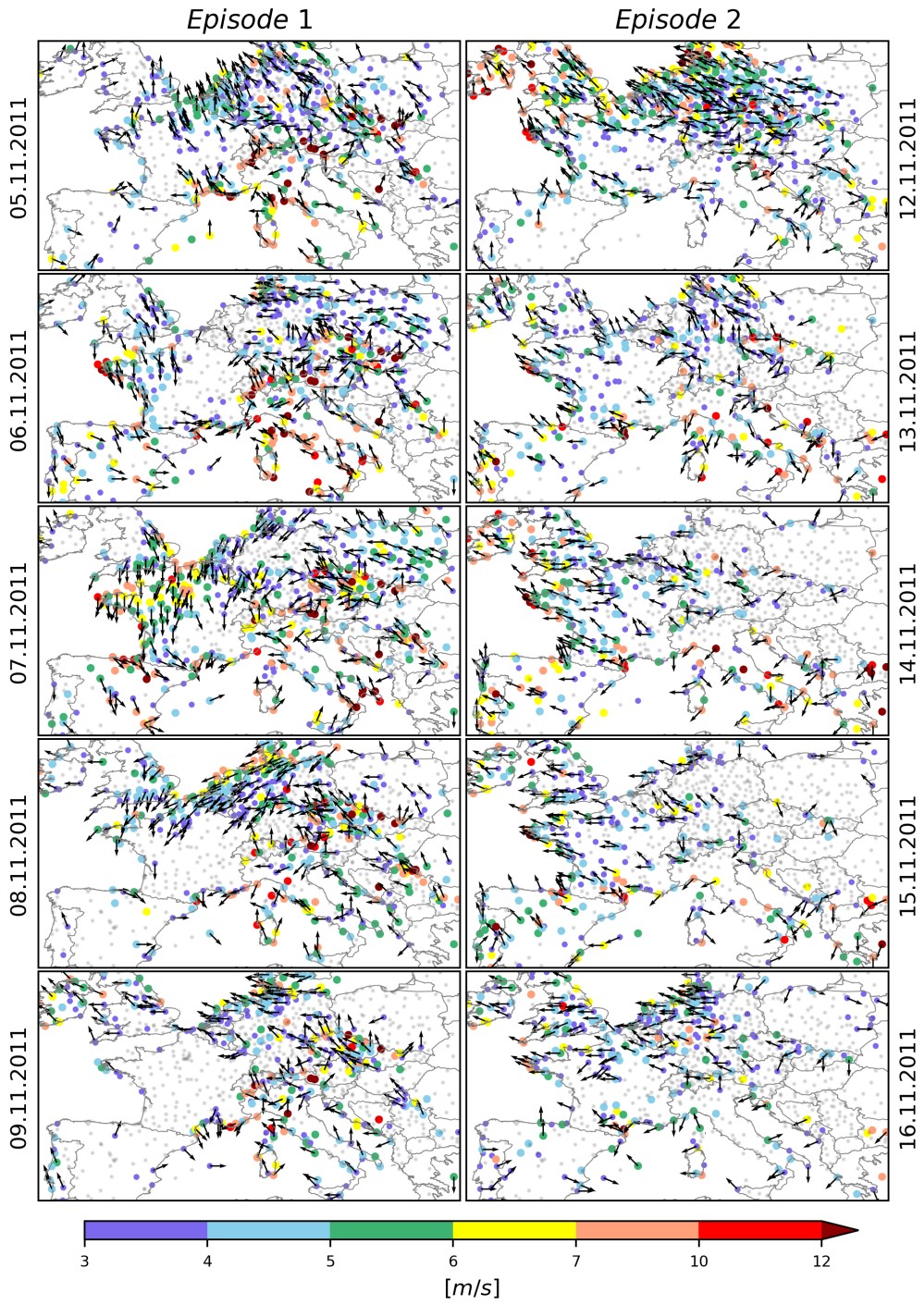

**Figure 5.** Daily averaged wind speed and directions during two high pollution episodes. Episode 1, from 5 to 9 November (episode 1, left) and from 12 to 16 November (episode 2, right).Stations with measured wind speed bellow 3 m/s are marked with grey dots.

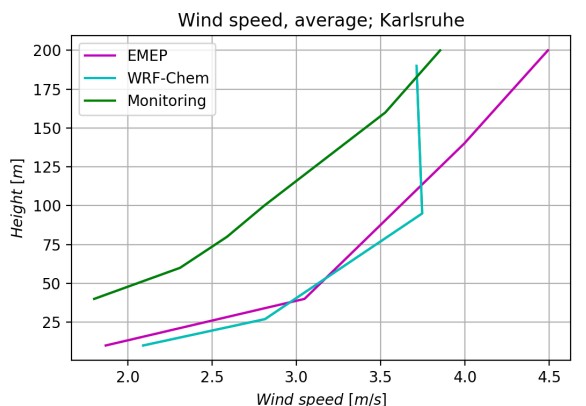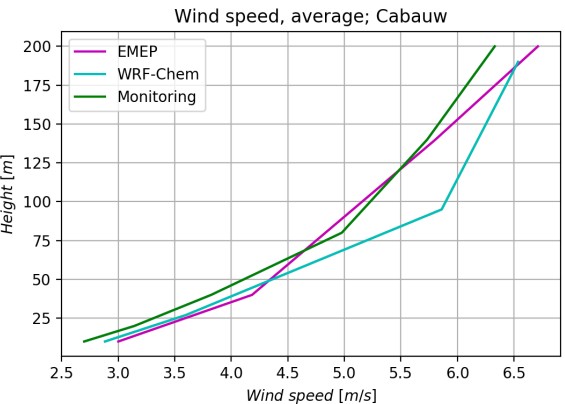

**Figure 6.** Vertical profiles of measured and modelled wind speeds at Karlsruhe (left, measurements source: Institute of Meteorology and Climate Research, Atmospheric Environmental Research, Karlsruhe Institute of Technology) and Cabauw mast station (left, measurements source: Cesar Observatory, http://www.cesar-observatory.nl) during November 2011.

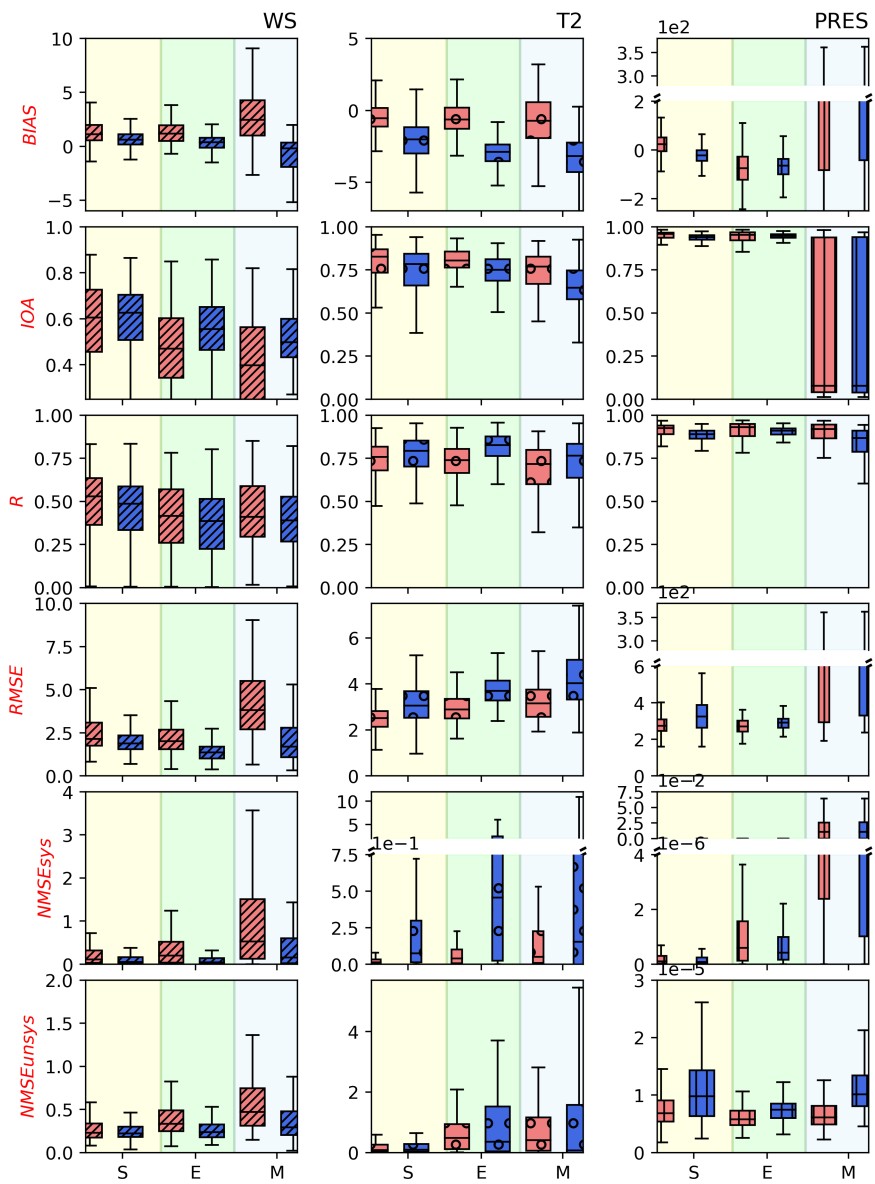

**Figure 7.** Intercomparison of the applied statistical measures (*BIAS*, *IOA*, *r*, *RMSE*, *NMSEsys*, *NMSEunsys*) between modelled (WRF-Chem – red boxes, EMEP – blue boxes) and measured (from 920 meteorological stations across all of Europe) wind speed (*//*), temperature (° °) and surface pressure (‖) during November 2011 for sea-level (S), elevated (E) and mountain (M) stations. The units of selected meteorological parameters are m/s for wind speed, °C for temperature and hPa for surface pressure

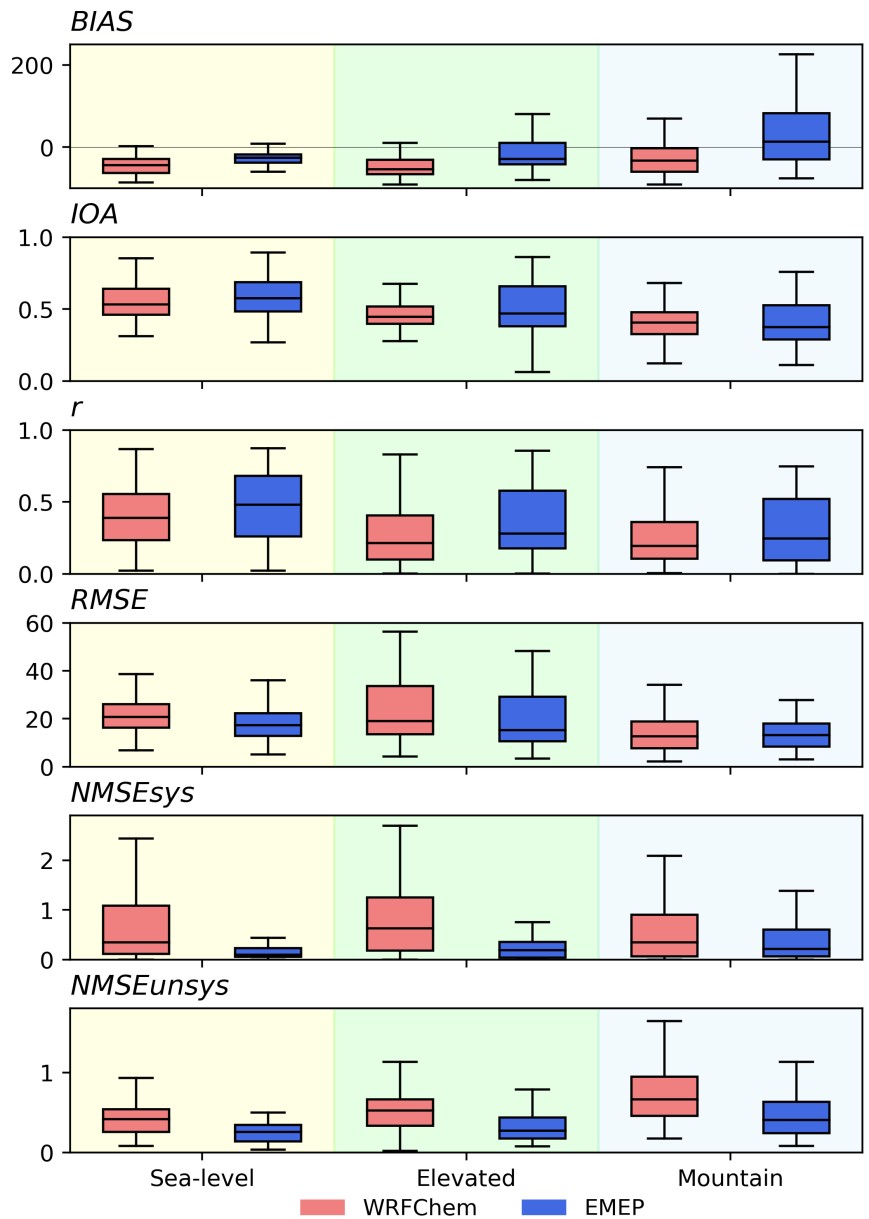

**Figure 8.** Intercomparison of the applied statistical measures (*BIAS, IOA, r, RMSE, NMSEsys, NMSEunsys*) between measured $\left(\overline{PM_{10}}\right)_d$ (310 rural background stations from Airbase, http://acm.eionet.europa.eu/databases/airbase and the EU-PHARE project) and modelled $\left(\overline{PM_{10}}\right)_d$ with the WRF-Chem (red boxes) and EMEP (blue boxes) models during November 2011 with respect to the station height.

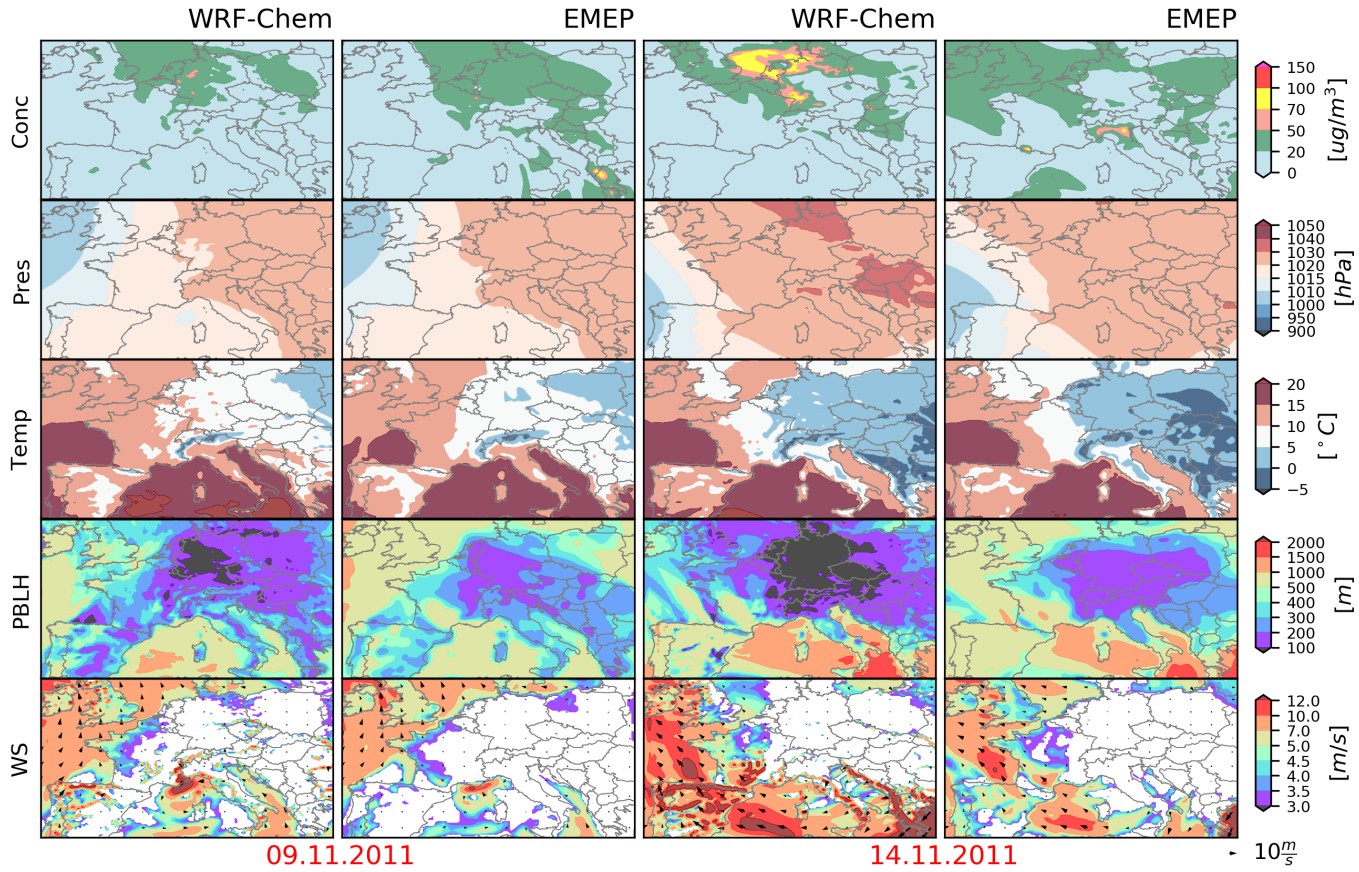

**Figure 9.** Modelled $\left(\overline{PM_{10}}\right)_d$ as *Conc*, and $\left(\overline{mslp}\right)_d$ as *Pressure*, $\left(\overline{t_{2m}}\right)_d$ as *Temp*, $\left(\overline{pblh}\right)_d$ as *PBLH* and $\left(\overline{ws}\right)_d$ with $\left(\overline{wd}\right)_d$ as *WS* for two typical days during the first (09 November 2011) and second (14 November 2011) high pollution episodes from the WRF-Chem and EMEP models, respectively.

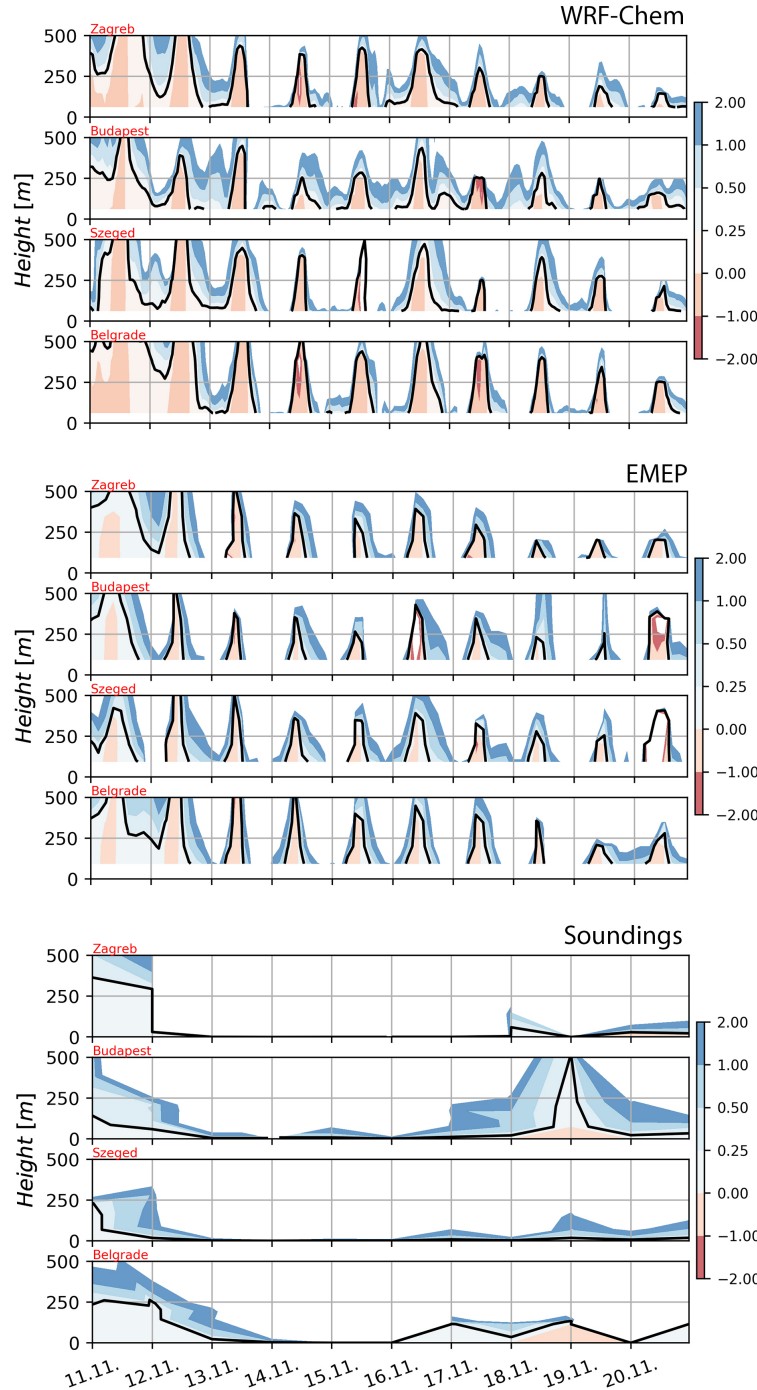

**Figure 10.** Time series of the vertical profile of the bulk Richardson number (equation 9, the colour bar on the right) for the Zagreb, Budapest, Szeged and Belgrade sites from WRF-Chem and EMEP model and sounding measurements before/after and during second pollution episode (from 11 to 21 November). The black line indicates the boundary layer height.

**Table 1.** The number of stations used in the analysis.

| Station altitude | Airbase stations | Meteorology stations |
|---|:---:|---:|
| Sea-level | 121 | 366 |
| Elevated | 107 | 335 |
| Mountain | 92 | 219 |

**Table 2.** Details of the WRF-Chem parameterizations.

| Parameterization | Used scheme |
|---|---|
| Microphysics | Lin et al. scheme |
| Long-wave radiation | rrtm scheme |
| Short-wave radiation | Goddard shortwave |
| Land surface model | Unified Noah land-surface model |
| Surface layer | Monin-Obukhov (Janjić) scheme |
| Boundary layer scheme | Mellor-Yamada-Janjić TKE scheme |
| Cumulus physics | Kain-Fritsch (new Eta) scheme |
| Gas-phase mechanism | RADM2 |
| Aerosol module | MADE/SORGAM (including some aqueous reactions) |
| Chemical initial conditions | From Mozart global model |
| Chemical boundary conditions | Idealized profile (from Mozart global model) |