# Peer review of "Regional-scale modelling for the assessment of atmospheric particulate matter concentrations at rural background locations in Europe"

_Atmospheric Chemistry and Physics, 2019_

## Referee Comment (RC1) · Anonymous Referee #1 · 12 Nov 2019

The manuscript deals with interesting topic of aerosol modelling and presents results of simulation of two modelling systems and high number of stations. The review recognizes that plenty of work has been done with processing of all data. On the other hand, many serious errors occurs in the manuscript and its current state absolutely does not respond to the ACP level. The most serious is the wrong use of statistical variables (see below), nearly no comparison with previous studies, sometime wrong or not described methodology. Also the presentation quality is not well, the text is hard to understand, sentences are often wrong arranged and mistakes in English occur (in/definite articles, commas, word order, braces). The number of technical errors is very height (see below). The manuscript have to be fundamentally improved or otherwise rejected.

Specific major comments:

1) Fig. 7: Application of the same Equation 1 for BIAS to wind speed, temperature and surface pressure is not a good idea. E.g., a small hPa BIAS have great consequences, but percentage BIAS is only slight. For temperature, it depends on a choice if temperature in Celsius or Kelvins is filled. Generally, these variables are evaluated usually by simple BIAS as only the difference between model and measured value. Similarly, also NMSE, NMSEsys, NMSEunsys are dependent on Celsius/Kelvins, therefore not appreciate for temperature evaluation. Further, in every case, it is not possible to compare used statistical variables for comparison between meteorological variables. For this reason, it is also necessary to modify sentences in p. 12/l. 8-22.

2) Chap. 3.3.1: In general, evaluation of EMEP meteorology means evaluation of IFS model, EMEP is only the chemical transport model. This should be taken into consideration and discussed.

3) P. 13/ l. 17-23: It is not reasonable to conclude that the overall performance of models was good, due to low correlation of PM10 concentrations and no comparison with other similar modelling studies. The comparison with other studies is relevant also for temperature and other meteorological variables evaluated. The comparison with previous studies has to be added to the paper.

4) Chap. 3.3.2: The text of the chapter is more a synoptic situation description than the model evaluation. There is no numeric comparison and model vs. observed spatial distributions of variables are in different figures, moreover partly with different scales. Please enable better comparison of modelled and observed values (figures including modelled and observed values, some statistics focused on the episodes).

5) Chap. 3.3.3: There is no information about sounding measurement, e.g. source, temporal resolution of data, etc. In case that only daily means are available, it is necessary to compare also only model daily means. White colour stands for RiB > 1 (not 0.25 as in text), statically stable conditions are for RiB > 0.25. It would be appropriate

to have a better colour scale clearly pronouncing the limit of RiB=0.25. Further, why did not used direct output of boundary layer height from models? And again, the modelled and observed data could be in the same figure to easier comparison and reduction of total number of figures in the paper.

6) Chap. 4: Comparison with other previous studies belongs rather to sections Discussion (or Results), not in section called Summary or Conclusions.

7) Chap. 3.2, p. 9/ l. 23-27: Weather in western Europe should be also described, due to region of above-average PM concentrations.

Other comments and technical corrections:

1) P. 2/ l. 33-34: WRF-Chem includes chemical reactions in gas-phase mechanism used.

2) AQMEII and EMEP could be referenced by citation.

3) P. 3/l. 29: Braces in braces.

4) 4/33: Uncertainties are calculated but not used, it is not necessary to write it.

5) Chap. 2.2: Please write the reason for using of specific statistical methods and what they describe (at least for less frequent ones)

6) 2.2: There are two mistakes in IOA definition (see e.g. https://www.rforge.net/doc/packages/hydroGOF/d.html)

7) 2.4: first paragraph belongs rather to introduction – sentence 6/16-18 is unclear.

8) 2.4.1: Which type of chemical mechanisms (gases, aerosols) is used in EMEP model?

9) 2.4.1: What horizontal resolution has IFS?

10) 2.4.1: Which PBL parametrization is used in terms of meteorological model?

11) 2.4.2: Why is not used the same domain for both models?

12) 2.4.2: Please add version of WRF-Chem model.

13) 2.4.2: Add the reason why is used NCEP analysis (resolution of 1 deg) for meteo-rological ICBC and not ERA-int reanalysis (0.7 deg).

14) 8/14: Which differences are meant? Description of results belongs to Chap. 3.

15) 8/18-24: Sentences in the whole paragraph are unclear and should be written better.

16) Chap. 2: There is not written any time extent of performed simulations or no information about spin-up interval.

17) 3.1 9/5: Analysis of variance should be shortly described or referenced. What does mean abbreviation ANOVA and p=0?

18) Fig. 2: The format of time axis (MM.DD.) is misleading due to fact that the paper concerns also to episodes. Someone can understand it as episode between January 1st and 12th. (Fig. 9+10 have time format DD.MM.)

19) 3.1: Secondary Inorganic Aerosols (SIA) – please reference it or describe more.

20) 10/8: There is maybe any missing text or reference to Fig. S2.

21) 10/23: Poland borders with eastern parts of Germany, so the onset could be rather in Poland and north-eastern Germany.

22) Fig. 3+4: Stations with temperature between 5-10 °C are not well visible.

23) Fig. 3+4: It seems that mountains stations indicate lower surface pressure, it would be appropriate to explain it.

24) 11/25: No significant difference between models and measurement below 75 m – that is not true (Fig. 6, on the left)

25) Fig. 9: Please explain WS.

26) 14/32-33: It is not evident from Fig. 4 that there were increased values of PM10 in Pannonian basin, only few stations occur in this area. It should be well discussed or not written.

27) Fig. 11: Obviously wrong description.

---

## Referee Comment (RC2) · Anonymous Referee #2 · 22 Dec 2019

Please revision the english language

Please review te conclusions They are confused. Conclusions should be done in short setences and with the main cnclusions each by each (in paragraph)
* * *

---

## Author Comment (AC2) · 2 Feb 2020

Please revision the english language

Please review the conclusions. They are confused. Conclusions should be done in short setences and with the main conclusions each by each (in paragraph)

The authors would like to thank Reviewer for constructive comment that improved the quality of our paper. We made a modification of conclusions. With these corrections, we are also attaching Proofreading confirmation. Please find corrected part of Summary and conclusions below (in purple colour):

[revised manuscript text omitted]

---

## Author Response (AR1)

The manuscript deals with interesting topic of aerosol modelling and presents results of simulation of two modelling systems and high number of stations. The review recognizes that plenty of work has been done with processing of all data. On the other hand, many serious errors occurs in the manuscript and its current state absolutely does not respond to the ACP level. The most serious is the wrong use of statistical variables (see below), nearly no comparison with previous studies, sometime wrong or not described. Also the presentation quality is not well, the text is hard to understand, sentences are often wrong arranged and mistakes in English occur (in/definite articles, commas, word order, braces). The number of technical errors is very height (see below). The manuscript have to be fundamentally improved or otherwise rejected.

Author response:

The authors would like to thank Reviewer for detailed and constructive comments that helped to improve our paper significantly. We have modified the manuscript accordingly and all comments were tackled. Please find our explanations bellow. We are attaching also Proofreading confirmation at the end of this pdf file. Please find answers in black colour and new or corrected parts of manuscript in purple colour. We made few more technical corrections in this answers from our last respond on 2 Feb 2020.

Specific major comments:

1) Fig. 7: Application of the same Equation 1 for BIAS to wind speed, temperature and surface pressure is not a good idea. E.g., a small hPa BIAS have great consequences, but percentage BIAS is only slight. For temperature, it depends on a choice if temperature in Celsius or Kelvins is filled. Generally, these variables are evaluated usually comment by simple BIAS as only the difference between model and measured value. Similarly, also NMSE, NMSEsys, NMSEunsys are dependent on Celsius/Kelvins, therefore not appreciate for temperature evaluation. Further, in every case, it is not possible to compare used statistical variables for comparison between meteorological variables. For this reason, it is also necessary to modify sentences in p. 12/l. 8-22.

Author response:

Figure 7 is updated with separated y axis (each parameters has its own plot) using new equations for IOA (proposed formula from hydro package of R) and BIAS (representing difference between the

model and measurements). With this approach, the applied statistics (statistic measure) on parameters can be compared regardless of magnitude of the measure itself (e.g., BIAS, RMSE, etc.). The main goal of Figure 7 was to show how the model reproduces meteorological conditions on particular level, and to analyse the relation between parameters (e.g., does the model tend to increase in performance with height for all parameters, or not). The units of meteorological parameters are now written in the Figure caption, and the dependence of statistic performance regarding units is described in the text. As a consequence of changing a IOA equation, Figure 8 is as well updated. Please find corrected text and new Figure 7 bellow and in Revised Manuscript p5, L15; p12 L31 - p13 L17),

2.2. Statistical analysis

[revised manuscript text omitted]

2) Chap. 3.3.1: In general, evaluation of EMEP meteorology means evaluation of IFS model, EMEP is only the chemical transport model. This should be taken into consideration and discussed.

The comment is accepted and included in the manuscript in a form similar as for WRF-Chem model (p8, L18-21 in originally submitted manuscript). The following change can be found at the end of 2.4.1 section. Corrected text in the revised manuscript on p8, L8-12

The above-written setup of EMEP model with the IFS meteorology as an initial and boundary meteorological conditions is later on referred and used in a form as "EMEP model". Any further comparison of meteorological conditions obtained in EMEP simulations is related to the IFS model and $PM_{10}$ to the choice of EMEP chemistry parameterization.

3) P. 13/ l. 17-23: It is not reasonable to conclude that the overall performance of models was good, due to low correlation of PM10 concentrations and no comparison with other similar modelling studies. The comparison with other studies is relevant also for temperature and other meteorological variables evaluated. The comparison with previous studies has to be added to the paper.

The comment is accepted and the comparisons with previous studies regarding modelling studies on meteorological and chemical parameters are added in the text (section 3.3.1). Corrected text in the manuscript, for chemistry (Revised manuscript, p14, L23-24)

The overall performance of the models regarding $\overline{\left(PM_{10}\right)}_d$ was good, and the results are in agreement with similar modeling studies (e.g, Werner et al., 2015; Baró et al., 2015; Forkel et al., 2015; Gauss et al., 2016)

..for meteorology (Revised manuscript, p13, L23-27)

Based on given statistic, overall model performance regarding meteorological parameters was in accordance to similar modeling studies. For example, negative BIAS and high r for $\overline{\left(t_{2m}\right)}_d$ was found in e.g. Skjøth et al., 2015, Qu et. al, 2014. Positive BIAS for $\overline{\left(ws\right)}_d$ was already addressed as an issue in related studies such as e.g. Baró et al., 2015; Forkel et al., 2015, while results for $\overline{\left(mslp\right)}_d$ for sea-level and/or elevated stations are in accordance with e.g. Qu et. al, 2014.

4) Chap. 3.3.2: The text of the chapter is more a synoptic situation description than the model evaluation. There is no numeric comparison and model vs. observed spatial distributions of variables

are in different figures, moreover partly with different scales. Please enable better comparison of modelled and observed values (figures including modelled and observed values, some statistics focused on the episodes).

Table with the observed and modelled values as well as statistics are now added in the Supplementary Information and discussed in this section. Corrected text in the revised manuscript p15, L27 –L12:

The *SI* Tables S1 – S2 are showing the minimum, maximum and median values of $\left(\overline{PM_{10}}\right)_d$, $\left(\overline{t_{2m}}\right)_d$, $\left(\overline{pblh}\right)_d$, $\left(\overline{mslp}\right)_d$, $\left(\overline{ws}\right)_d$ over the domain (Fig 1) for both models during episodes. Minimum, maximum and median values of $\left(\overline{mslp}\right)_d$ between models were similar. Average minimum $\left(\overline{mslp}\right)_d$ over domain was 1004.77 hPa and 1005.55 hPa, average maximum 1031.93 hPa and 1031.44 hPa and average median 1021.18 hPa and 1020.33 hPa for WRF-Chem and EMEP model respectively. The average minimum $\left(\overline{t_{2m}}\right)_d$ for WRF-Chem ~-5.54◦C was lower in respect to EMEP model ~-2.31◦C, however average maximum $\left(\overline{t_{2m}}\right)_d$~20◦C and median $\left(\overline{t_{2m}}\right)_d$~10◦C values were same for both models. $\left(\overline{pblh}\right)_d$ in WRF-Chem model varied from an average minimum value of 38.97 m to an average maximum value of 1612.29 m, while EMEP had much higher average minimum value 137.62 m (due to coarser vertical resolution of the EMEP model) and somewhat lower average maximum value ~ 1585.81 m (*SI* Tables S1 – S2). $\left(\overline{ws}\right)_d$ was more variable over the domain for WRF-Chem in respect to the EMEP model. During both episodes, minimum $\left(\overline{ws}\right)_d$ in WRF-Chem was in the range from 0 to 0.11 m/s, while maximum varied from 19.77 m/s up to 36.34 m/s, the average median $\left(\overline{ws}\right)_d$ was 5.00 m/s. For EMEP model, minimum $\left(\overline{ws}\right)_d$ was similar to WRF-Chem, and in the range from 0.01 m/s to 0.18 m/s, while maximum $\left(\overline{ws}\right)_d$ was lower than obtained with WRF-Chem simulation, in the range from 12.74 m/s to 16.77 m/s. Same was as well as for the average median $\left(\overline{ws}\right)_d$ lower than in WRF-Chem simulation, 3.60 m/s. The average $\left(\overline{PM_{10}}\right)_d$ concentrations were generally higher in the EMEP model. The average minimum $\left(\overline{PM_{10}}\right)_d$ concentrations were between 0.19 and 1.51 µg/m$^3$, average maximum $\left(\overline{PM_{10}}\right)_d$ was 62.04 µg/m$^3$ and 84.45 µg/m$^3$ and average median $\left(\overline{PM_{10}}\right)_d$ values were between 6.91 µg/m$^3$ and 13.46 µg/m$^3$ for WRF-Chem and EMEP model respectively during both episodes. The absolute maximum concentration obtained with the WRF-Chem model was 63.55 µg/m$^3$ and 81.32 µg/m$^3$ while for the EMEP model, 110.09 µg/m$^3$ and 97.84 µg/m$^3$ during the first and second episode, respectively.

**Table S2.** The minimum, maximum and median values of $\overline{(PM_{10})}_d$, $\overline{(mslp)}_d$, $\overline{(t_{2m})}_d$, $\overline{(pblh)}_d$, $\overline{(ws)}_d$ over the domain (Fig 1) for WRF-Chem model during both episodes.

| | $\overline{(PM_{10})}_d$ | | | $\overline{(mslp)}_d$ | | | $\overline{(t_{2m})}_d$ | | | $\overline{(pblh)}_d$ | | | $\overline{(ws)}_d$ | | |
| Day | MIN | MAX | MEDIAN | MIN | MAX | MEDIAN | MIN | MAX | MEDIAN | MIN | MAX | MEDIAN | MIN | MAX | MEDIAN |
|---|---|---|---|---|---|---|---|---|---|---|---|---|---|---|---|
| 5 | 0.17 | 34.76 | 4.17 | 998.60 | 1,025.76 | 1,011.24 | 0.78 | 21.81 | 11.93 | 51.14 | 1,581.14 | 404.68 | 0.03 | 36.34 | 5.59 |
| 6 | 0.18 | 63.55 | 3.83 | 999.13 | 1,031.99 | 1,018.62 | -0.46 | 21.48 | 11.48 | 45.80 | 1,633.84 | 419.54 | 0.11 | 34.02 | 6.72 |
| 7 | 0.19 | 47.24 | 5.39 | 1,004.90 | 1,033.89 | 1,019.55 | -2.01 | 21.40 | 10.65 | 46.06 | 1,616.29 | 319.23 | 0.09 | 25.03 | 5.38 |
| 8 | 0.19 | 57.69 | 8.15 | 1,006.33 | 1,030.76 | 1,017.09 | -1.11 | 21.31 | 10.74 | 34.41 | 1,450.01 | 297.31 | 0.04 | 27.08 | 4.52 |
| 9 | 0.19 | 59.39 | 7.74 | 1,001.21 | 1,028.03 | 1,018.95 | -4.70 | 21.27 | 11.36 | 41.57 | 1,250.24 | 284.93 | 0.02 | 19.89 | 3.87 |
| 12 | 0.19 | 58.43 | 6.98 | 1,008.10 | 1,040.86 | 1,027.97 | -9.75 | 20.69 | 11.02 | 40.80 | 1,525.72 | 330.18 | 0.06 | 25.97 | 5.66 |
| 13 | 0.19 | 81.32 | 7.05 | 1,007.63 | 1,038.40 | 1,030.70 | -8.80 | 21.11 | 10.74 | 27.44 | 1,899.58 | 299.69 | 0.01 | 27.17 | 5.32 |
| 14 | 0.20 | 81.05 | 8.12 | 1,005.11 | 1,031.63 | 1,026.08 | -8.94 | 20.67 | 9.24 | 26.38 | 1,955.07 | 260.41 | 0.01 | 26.52 | 4.70 |
| 15 | 0.19 | 70.83 | 9.37 | 1,007.43 | 1,029.09 | 1,021.05 | -10.03 | 19.87 | 8.39 | 29.37 | 1,708.43 | 299.64 | 0.00 | 21.72 | 4.39 |
| 16 | 0.20 | 66.17 | 8.29 | 1,009.28 | 1,028.93 | 1,020.55 | -10.33 | 19.28 | 7.84 | 39.76 | 1,502.60 | 305.99 | 0.02 | 19.77 | 3.89 |
| AVG | 0.19 | 62.04 | 6.91 | 1,004.77 | 1,031.93 | 1,021.18 | -5.54 | 20.89 | 10.34 | 38.27 | 1,612.29 | 322.16 | 0.04 | 26.35 | 5.00 |

**Table S3.** The minimum, maximum and median values of $\overline{(PM_{10})}_d$, $\overline{(mslp)}_d$, $\overline{(t_{2m})}_d$, $\overline{(pblh)}_d$, $\overline{(ws)}_d$ over the domain (Fig 1) for EMEP model during both episodes.

| | $\overline{(PM_{10})}_d$ | | | $\overline{(mslp)}_d$ | | | $\overline{(t_{2m})}_d$ | | | $\overline{(pblh)}_d$ | | | $\overline{(ws)}_d$ | | |
| Day | MIN | MAX | MEDIAN | MIN | MAX | MEDIAN | MIN | MAX | MEDIAN | MIN | MAX | MEDIAN | MIN | MAX | MEDIAN |
|---|---|---|---|---|---|---|---|---|---|---|---|---|---|---|---|
| 5 | 1.10 | 82.55 | 12.78 | 1,000.20 | 1,026.08 | 1,011.79 | 0.27 | 20.82 | 11.42 | 134.42 | 1,628.13 | 614.25 | 0.04 | 14.81 | 4.64 |
| 6 | 1.13 | 110.09 | 11.76 | 1,003.34 | 1,030.98 | 1,018.15 | 0.31 | 21.13 | 10.94 | 132.20 | 1,604.03 | 609.60 | 0.18 | 16.11 | 4.99 |
| 7 | 1.19 | 95.88 | 12.37 | 1,006.36 | 1,033.38 | 1,019.29 | -0.82 | 20.72 | 10.15 | 153.99 | 1,406.29 | 499.97 | 0.04 | 15.70 | 3.58 |
| 8 | 1.16 | 73.12 | 14.54 | 1,005.62 | 1,029.38 | 1,017.84 | -0.63 | 20.44 | 10.43 | 137.09 | 1,244.23 | 444.85 | 0.03 | 14.09 | 3.10 |
| 9 | 1.25 | 78.60 | 12.54 | 1,001.58 | 1,028.91 | 1,019.23 | -1.35 | 20.22 | 11.28 | 152.47 | 1,163.56 | 392.36 | 0.04 | 12.74 | 2.63 |
| 12 | 2.64 | 81.96 | 14.91 | 1,007.41 | 1,040.12 | 1,026.67 | -3.68 | 20.02 | 11.36 | 122.74 | 1,732.04 | 503.42 | 0.04 | 15.24 | 3.67 |
| 13 | 2.40 | 72.48 | 15.43 | 1,008.17 | 1,038.18 | 1,028.08 | -4.57 | 19.74 | 10.95 | 122.30 | 1,843.41 | 495.59 | 0.01 | 17.28 | 4.08 |
| 14 | 1.61 | 97.84 | 15.35 | 1,005.91 | 1,030.13 | 1,023.16 | -4.76 | 20.01 | 9.43 | 126.60 | 1,778.65 | 503.42 | 0.05 | 16.77 | 3.66 |
| 15 | 1.23 | 72.21 | 12.65 | 1,008.09 | 1,028.25 | 1,018.74 | -3.31 | 19.22 | 8.34 | 147.04 | 1,798.95 | 487.96 | 0.03 | 14.32 | 3.08 |
| 16 | 1.36 | 79.80 | 12.36 | 1,008.81 | 1,029.01 | 1,020.32 | -4.58 | 19.27 | 8.44 | 147.33 | 1,658.80 | 423.41 | 0.02 | 11.86 | 2.58 |
| AVG | 1.51 | 84.45 | 13.47 | 1005.55 | 1031.44 | 1020.33 | -2.31 | 20.16 | 10.27 | 137.62 | 1585.81 | 497.48 | 0.05 | 14.89 | 3.60 |

5) Chap. 3.3.3: There is no information about sounding measurement, e.g. source, temporal resolution of data, etc. In case that only daily means are available, it is necessary to compare also only model daily means. White colour stands for RiB > 1 (not 0.25 as in text), statically stable conditions are for RiB > 0.25. It would be appropriate to have a better colour scale clearly pronouncing the limit of

RiB=0.25. Further, why did not used direct output of boundary layer height from models? And again, the modelled and observed data could be in the same figure to easier comparison and reduction of total number of figures in the paper.

We have provided more information on temporal resolution, time step comparison in the manuscript (Section 3.3.3). Soundings were available at 00 and 12 UTC (not daily average) and only corresponding model vertical profiles were used for the intercomparison. The colour scale is updated, the Figures are placed in the same plot in order to reduce total number of figures in the text. Corrected text in the revised manuscript p7, L4-L7; p16, L32 – p17, L5:

2.3 Boundary layer height determination

Comparison of estimated planetary boundary layer height (PBLH) was carried out using equation (9) rather than comparing the direct output of model-derived PBLH values as each model is using a different method for calculation of the PBLH. By using the same methodology for PBLH determination, uncertainties are reduced and the more realistic evaluation of two modelled PBLH values is assured.

3.3.3  Intercomparison of modelled PBL height against radio soundings

It must be pointed out that available sounding measurements were instantaneous values at 00 UTC only, while time step in WRF-Chem model was 1 hour and in EMEP 3h. The $Ri_B$ values calculated from soundings and modeled data shown on Fig 10 are represented with the same time step as input data: 12h for measurements, 1 h for WRF-Chem and 3h for EMEP model. According to Fig 10, the models were consistent in $Ri_B$ and in estimating $H_{bl}$. The development of the atmospheric boundary layer started early in the morning with sunrise and reached values up to 350 – 400 m around 14:00 (local time), except between 17 and 21 November when a decrease in $H_{bl}$ was found. During this period the peak values of $H_{bl}$ reached 200 m and the statically stable conditions ($Ri_B$ >0.25) were dominant (light blue to dark blue color up to value of 2, above in white colour).

[Figure]

Figure 10. Time series of the vertical profile of the bulk Richardson number (equation 8, the colour bar on the right) for the Zagreb, Budapest, Szeged and Belgrade sites from WRF-Chem and EMEP model

and sounding measurements before/after and during second pollution episode (from 11 to 21 November). The black line indicates the boundary layer height.

6) Chap. 4: Comparison with other previous studies belongs rather to sections Discussion (or Results), not in section called Summary or Conclusions.

Comment is accepted. However not all comparisons with previous studies were moved as we think that the comparison with previous activities of AQMEII group is important in order to clearly explain the contribution of our work in relation to other modelling studies. It must be point out that we have as well completely rearranged the Summary and conclusions chapter due to Reviewer 2 comment. The following sentence was moved to Introduction, revised manuscript, p2, L33-35:

Other studies (e.g., Saide et al., 2011) also indicated challenges in the modelling of PM mass, especially during statically stable atmospheric conditions, due to the choice of vertical and horizontal resolution as well as the influence of vertical and horizontal diffusion coefficients during model setup (Jeričević et al., 2010).

7) Chap. 3.2, p. 9/ l. 23-27: Weather in western Europe should be also described, due to region of above-average PM concentrations.

The comment is accepted. Description of weather conditions for Western Europe is added. Here is an added text in the revised manuscript p10, L13-19:

In Western Europe, the autumn season temperature was above average normal (1961-1990) and was characterized by prevailing high-pressure field. This was observed particularly in November during which monthly average temperature records were exceeded (e.g. UK, France and Switzerland reported their second 15 warmest autumn in last 100 years). Contrary to the Western Europe, the increased nocturnal cooling decreased temperatures in Southeastern Europe. The dominating high-pressure field resulted in a decrease of precipitation in some Western and Central Europe countries, e.g. south France, Alpine region, Germany, Austria, Czech Republic, Slovakia, Hungary. All those countries reported the driest November in more than the last 100 years (Blunden et al., 2012)

Other comments and technical corrections:
1)      P. 2/ l. 33-34: WRF-Chem includes chemical reactions in gas-phase mechanism used.
Comment accepted, correction was done. This part of text was deleted.

2)      AQMEII and EMEP could be referenced by citation.

Comment accepted, correction was done. They are now cited, revised manuscript p3, L2, L6.

3)      P. 3/l. 29: Braces in braces.

Comment accepted, unnecessary braces are deleted.

4)      4/33: Uncertainties are calculated but not used, it is not necessary to write it.

Comment accepted, this part is removed from manuscript.

5)      Chap. 2.2: Please write the reason for using of specific statistical methods and what they describe (at least for less frequent ones)

Comment accepted, description was added, the following text is added in the revised manuscript p6, L3-L20:

As there is no single best modelling performance measure, it is recommended by Chang and Hanna (2004) that a suite of different performance measures needs to be applied. Results should be carefully interpreted by taking into account advantages and disadvantages of all applied statistical measures and assuring that those are complementary to each other and leading to the same conclusion on the certain ability of the model performance. Therefore as already previously noted in this Section, a set of different statistical measures is used in order to understand the ability of the model to properly estimate high pollution episodes of $PM_{10}$ concentrations and to evaluate the relations between chemical and meteorological parameters. BIAS refers to the arithmetic difference between M and O indicating model's general overestimation or underestimation of analysed parameters. It is known that a model whose predictions are completely out of phase with observations to still have a BIAS = 0 because of compensating errors. Different BIAS was used: for evaluating model performance regarding $PM_{10}$ we used BIAS under equation (1) as opposed to meteorological parameters under equation (2). $r$ and IOA are dimensionless measure of model accuracy. $r$ is sensitive to a good agreement of extreme data pairs and a scatter plot might show generally poor agreement but the presence of a good agreement for a few extreme pairs will greatly improve $r$. The IOA is the ratio of the mean square error and the potential error and then subtracted from one (Willmott, 1984). The IOA varies from 0 to 1 with higher index values indicating that M have better agreement with the O. Although the IOA provides some improvement over the $r$, it is still sensitive to extreme values due to the square differences in the mean

square error in the numerator. *RMSE* gives information on the spread of the residuals from the regression line, it highly depends on the magnitude of the parameter on which *RMSE* is applied and therefore it cannot be compared with *RMSE* of some other parameter. $NMSE_{sys}$ is a measure which with $NMSE_{unsys}$ provide information on systematic and unsystematic (random) errors in the model.

6) 2.2: There are two mistakes in IOA definition (see e.g. https://www.rforge.net/doc/packages/hydroGOF/d.html)

Comment accepted. We changed the equation with the new proposed one. Thank you for noticing this bug. As a consequence, Figures 7, 8 and Table SI 1 were updated with new data and properly discussed in the text. Figure 7 and 8 are already inserted during Major 1 answer, please find bellow new equation for *IOA* and updated Table S1.

$$IOA = 1 - \frac{\sum_{i=1}^{N}(O_i - M_i)^2}{\sum_{i=1}^{N}\left(abs(M_i - \overline{O}) + abs(O_i - \overline{O})\right)^2}$$

**Table S1.** Intercomparison of applied statistical measures (BIAS, IOA, $r$, RMSE, $NMSE_{sys}$, $NMSE_{unsys}$) with minimum, median and maximum values, between measured $\left(\overline{PM_{10}}\right)_d$ (310 rural background stations from Airbase) and modelled $\left(\overline{PM_{10}}\right)_d$ with the WRF-Chem and EMEP models during November 2011 with respect to the station height (same as Fig 8).

| | | WRF-Chem | | | EMEP | | |
|---|---|---|---|---|---|---|---|
| | Height | MIN | MEDIAN | MAX | MIN | MEDIAN | MAX |
| BIAS | Sea-level | -86 | -44 | 2 | -68 | -26 | 47 |
| | Elevated | -91 | -55 | 100 | -80 | -29 | 132 |
| | Mountain | -91 | -33 | 196 | -76 | 13 | 226 |
| IOA | Sea-level | 0.3 | 0.5 | 0.9 | 0.3 | 0.6 | 0.9 |
| | Elevated | 0.2 | 0.4 | 0.9 | 0.1 | 0.5 | 0.9 |
| | Mountain | 0.1 | 0.4 | 0.9 | 0.1 | 0.4 | 0.8 |
| $r$ | Sea-level | 0.02 | 0.39 | 0.87 | 0.02 | 0.48 | 0.87 |
| | Elevated | 0.00 | 0.21 | 0.88 | 0.00 | 0.28 | 0.85 |
| | Mountain | 0.01 | 0.19 | 0.82 | 0.00 | 0.24 | 0.75 |
| RMSE | Sea-level | 6.9 | 20.7 | 60.8 | 5.0 | 17.3 | 50.2 |
| | Elevated | 4.2 | 19.6 | 114.7 | 3.5 | 15.8 | 111.0 |
| | Mountain | 2.2 | 12.7 | 36.6 | 3.0 | 13.2 | 34.0 |
| $NMSE_{sys}$ | Sea-level | 0.0 | 0.3 | 5.5 | 0.0 | 0.1 | 1.4 |
| | Elevated | 0.0 | 0.7 | 9.3 | 0.0 | 0.2 | 3.3 |
| | Mountain | 0.0 | 0.3 | 9.4 | 0.0 | 0.2 | 2.4 |
| $NMSE_{unsys}$ | Sea-level | -0.7 | 0.4 | 1.7 | -0.3 | 0.3 | 0.9 |

| | | | | | | |
|---|---|---|---|---|---|---|
| Elevated | 0.0 | 0.5 | 1.9 | 0.1 | 0.3 | 1.6 |
| Mountain | 0.2 | 0.7 | 2.4 | 0.1 | 0.4 | 1.5 |

6)      2.4: first paragraph belongs rather to introduction – sentence 6/16-18 is unclear.

Comment accepted. The paragraph is moved to the Introduction and addressed sentence was rephrased, revised manuscript p3, L11-L15. Below is rephrased sentence:

The offline models consider solving separately meteorological conditions prior to chemistry during the simulation runs. There exists a huge variety of offline models such as the Comprehensive Air Quality Model with Extensions, CAMx (EVIRON, 2010), the Community Multi-scale Air Quality, CMAQ (U.S. Environmental Protection Agency), EMEP and LOTOS-EUROS (e.g., Solazzo et al., 2012).

8) 2.4.1:      Which type of chemical mechanisms (gases, aerosols) is used in EMEP model?

We used default EMEP setup which is mainly described in Simpson et al. 2012., the following comment is added in the text at the end of Section 2.4.1, revised manuscript p8, L6-L7:

Other mechanism used in this work (e.g. chemical scheme: EmChem09, chemical preprocessor: GenChem) are described in Simpson et al., 2012.

9) 2.4.1: What horizontal resolution has IFS?

ECMWF IFS has 0.22 deg horizontal resolution.

10) 2.4.1: Which PBL parametrization is used in terms of meteorological model?

Entire WRF-Chem model setup is written in Table 2. We used Mellor Yamada Janić scheme in WRF-Chem and for EMEP Boundary layer scheme with changes in turbulence parameterization (details in Jeričević et al., 2010).

11) 2.4.2: Why is not used the same domain for both models?

The EMEP model has its own domain which covers all of Europe, this could not be changed. Due to computational demands, we extended WRF-Chem domain as much as possible to match EMEP domain. Although we are aware that the difference of domain coverage can have influence on results as a consequence of different boundary conditions (e.g. North Africa is not included in WRF-Chem simulation), we can say that for the purpose of this study the domains of models were satisfying. Before making serious simulations, we made sensitivity tests with different domains and setups for

WRF-Chem and the domain included in this paper gave us the best ratio of computing demands and quality of results.

12) 2.4.2: Please add version of WRF-Chem model.

Comment accepted, the WRF-Chem version is added in the text, revised manuscript p8, L17-18:

In this paper, we used the WRF-Chem version 3.5.1.

13) 2.4.2: Add the reason why is used NCEP analysis (resolution of 1 deg) for meteorological ICBC and not ERA-int reanalysis (0.7 deg).

Comment accepted. We add an explanation in the revised manuscript p8, L21-L24:

Initial and boundary meteorological conditions were provided by NCEP (National Centers for Environmental Prediction) Final Analysis (FNL ds083.2) with 1 degree of horizontal resolution and a time step of every 6 hours. They were selected based on previous research and other conducted studies with WRF or WRF-Chem model (e.g. Gašparac et al., 2016; Grgurić et al., 2013; Jeričević et al., 2017; Syrakov et al., 2016).

14) 8/14: Which differences are meant? Description of results belongs to Chap. 3.

Comment accepted. The sentences were moved to section 3.3. We added extra description, so they are now much clearer. With "differences" we meant the differences between used emission databases.

We have moved this part of text from Section 2.4.2 to revised manuscript p12, L8-L11:

It is worth noting that differences between used emission databases were found in the spatial variability of $PM_{10}$ emissions and in the gridded input emission fields above the entire domains of EMEP and WRF-Chem. Notable differences in emissions were found over the coastal areas and Eastern part of the domain particularly over Bosnia and Herzegovina, Serbia and Hungary which are crucial for the case studies analysed here.

..and place it before unclear one:

Aside from this, the difference in vertical resolution (first model level height – EMEP at 46 m, WRF-Chem at 22 m) can have a strong impact on surface concentrations and thus can be related to the differences in surface $PM_{10}$ concentrations obtained from the two used models.

15) 8/18-24: Sentences in the whole paragraph are unclear and should be written better.

Comment accepted, the paragraph is corrected, revised manuscript p8, L32 – p9, L35:

It is worth pointing out that the results of statistical analysis and model evaluation further on in the text will not describe the performance of the model itself, but rather will describe the performance of a set of selected parameterisations and chemical and meteorological initial and boundary conditions used in WRF-Chem model. Following this, when referring to the "WRF-Chem model" in the text, the authors are referring to the WRF-Chem model with the above-described setup (Table 2). The WRF-Chem simulation is performed from 29 October to 30 November and EMEP from 1 October to 30 November. As all statistical analysis was done for dates after 1 November the simulation length was long enough to overcome the effects of spin up time.

16) Chap. 2: There is not written any time extent of performed simulations or no information about spin-up interval.

Comment accepted, the following information is added in the revised manuscript p9, L1-L3:

The WRF-Chem simulation is performed from 29 October to 30 November and EMEP from 1 October to 30 November. As all statistical analysis was done for dates after 1 November the simulation length was long enough to overcome the effects of spin up time.

17) 3.1 9/5: Analysis of variance should be shortly described or referenced. What does mean abbreviation ANOVA and p=0?

Comment accepted, the following text is added in the revised manuscript p9, L24-L27:

The applied ANOVA is calculated via *scipy* python package. This particular one-way ANOVA tests the null hypothesis that two or more groups have the same population mean. The *p* value is common variable used in hypothesis testing, the smaller the p value, the stronger is the evidence that hypothesis needs to be rejected (Heiman et al., 2001).

18) Fig. 2: The format of time axis (MM.DD.) is misleading due to fact that the paper concerns also to episodes. Someone can understand it as episode between January 1st and 12th. (Fig. 9+10 have time format DD.MM.)

The comment accepted, the format of the time axis in Fig 2 is changed and now it is the same as one on Fig 9, 10.

[Figure]

Figure 2. The spatially averaged (upper panel) over all the rural background stations (the green line, corresponding to the right green y-axis) and the maximum of $\left(\overline{PM_{10}}\right)_d$ for all rural background stations (the red line, corresponding to the left red y-axis) and $\left(\overline{PM_{10}}\right)_d$ (lower panel) during 2011. The values above 50 $\mu$g/m$^3$ (red colour) represent values above the daily limit values for PM$_{10}$ under the 2008/50/EC

19) 3.1: Secondary Inorganic Aerosols (SIA) – please reference it or describe more.

The entire sentence was referenced by two references. We rearrange the sentence, now this is much clearer. Here is a rearranged paragraph in the revised manuscript p9, L8-L10:

Moreover, according to e.g., EEA, 2013, Saarikoski et al., 2008, aside from the primary sources (natural and anthropogenic), the secondary inorganic aerosols (SIA) and secondary organic aerosols (SOA) vary substantially across Europe from season to season, which indicates the presence of various PM$_{10}$ sources.

20) 10/8: There is maybe any missing text or reference to Fig. S2.

The SI Fig 2 is referenced in the Sec. 3, Section 3.2, in the p10, L4 in originally submitted manuscript and in revised manuscript p10, L23.

21) 10/23: Poland borders with eastern parts of Germany, so the onset could be rather in Poland and north-eastern Germany.

Comment accepted, text is corrected in the revised manuscript p11, L8-L10:

The onset of the event was in Poland and Northeastern Germany and encompassed the coastal areas of Northern Europe, the Benelux countries and Northern France in the following days until 9 November.

22) Fig. 3+4: Stations with temperature between 5-10 °C are not well visible.

Comment accepted. The main idea in these Figures was to make difference between warmer and colder areas of the Europe. We added a little dot for those points. Please find new Figures below:

[Figure]

Figure 3. *DF* and measurements from synoptic stations (relative humidity (Rel Hum), ambient temperature at 2m (Temp), and surface pressure) from the National Center for Environmental Prediction, Final Analysis (ds083.2) data during the first large-scale episode (5 to 9 November). Stations with a temperature between 0-5°C are marked with little grey dots due to better representativeness on the map.

[Figure]

Figure 4. Same as Fig 4, but during the second large-scale episode (12 to 16 November).

23) Fig. 3+4: It seems that mountains stations indicate lower surface pressure, it would be appropriate to explain it.

Comment accepted. Description was added at the end of 3.2 Section in the revised manuscript p11, L35 – p12, L2:

According to Figs. 3 – 4, during both episodes, mainly on all higher mountain stations within domain, the low- $\overline{(mslp)}_d$ was 35 observed. The $\overline{(mslp)}_d$ values were around 900 hPa which is common $\overline{(mslp)}_d$ for altitudes above 500 m. This means that in both cyclonic and anticyclonic conditions, the $\overline{(mslp)}_d$ was not disturbed and all processes such as advection, due to strong $\overline{(mslp)}_d$ gradients occurred mainly for sea-level and elevated stations

24) 11/25: No significant difference between models and measurement below 75 m – that is not true (Fig. 6, on the left)

Comment accepted. The sentence is corrected in the revised manuscript p11, L18-L19:

During November there was no significant difference between modelled vertical profiles of wind speed below 75 m (Fig 6) for both sites.

25) Fig. 9: Please explain WS.

Comment accepted. It was already written in the caption that WS represents wind speed with direction. We added extra information in parentheses:

Figure 9. Modelled as *Conc*, and as *Pressure*, as $\left(\overline{t_{2m}}\right)_d$ *Temp*, $\left(\overline{pblh}\right)_d$ as *PBLH* and $\left(\overline{ws}\right)_d$ with $\left(\overline{wd}\right)_d$ as WS (wind speed and wind direction with color bar representing magnitude of wind speed) for two typical days during the first (09 November 2011) and second (14 November 2011) high pollution episodes from the WRF-Chem and EMEP models, respectively.

26) 14/32-33: It is not evident from Fig. 4 that there were increased values of PM10 in Pannonian basin, only few stations occur in this area. It should be well discussed or not written.

Comment accepted, the following description was added, revised manuscript p16, L23-27:

Pannonian basin endured high pollution events during the second high pollution episode that were mainly found at urban stations (not shown) due to the lack of rural background measurements. In the analysed period, increased values of $\left(\overline{PM_{10}}\right)_d$ can be depicted only on one available rural background station in the area, Fig 4. The increased concentrations can be observed also from modelling results (Fig 9, *SI* Fig S2). The area of increased concentrations is in accordance with the area of weak wind conditions (Fig 5) and low $\left(\overline{pblh}\right)_d$ values and can be described as an area of potentially statically stable conditions.

27) Fig. 11: Obviously wrong description.

Comment accepted. However, due to Major 5 comment, Figures 9-11 were merged into one, so there is no longer Fig 11 caption in the text.

**Anonymous Referee #2**

Please revision the english language

Please review the conclusions. They are confused. Conclusions should be done in short

setences and with the main conclusions each by each (in paragraph)

The authors would like to thank Reviewer for constructive comment that improved the quality of our paper. Please find modified conclusions bellow (in purple colour) as well as in revised manuscript on p17 L12 – p18 30. With these corrections, we are also attaching Proofreading confirmation at the end of this pdf file. We made a few more technical corrections in respect to one from our last respond (2 February).

[revised manuscript text omitted]

**EDITORIAL CERTIFICATE**

This document certifies that the manuscript listed below was edited for proper English language, grammar, punctuation, spelling, and overall style by one or more of the highly qualified native English speaking editors at American Journal Experts.

**Manuscript title:**

Regional-scale modelling for the assessment of atmospheric particulate matter concentrations at rural background locations in Europe

**Authors:**

Gašparac Goran, Jeričević Amela, Prashant Kumar, Branko Grisogono

**Date Issued:**

April 1, 2019

**Certificate Verification Key:**

21D4-2086-C842-302A-784E

[Figure]

This certificate may be verified at www.aje.com/certificate. This document certifies that the manuscript listed above was edited for proper English language, grammar, punctuation, spelling, and overall style by one or more of the highly qualified native English speaking editors at American Journal Experts. Neither the research content nor the authors' intentions were altered in any way during the editing process. Documents receiving this certification should be English-ready for publication; however, the author has the ability to accept or reject our suggestions and changes. To verify the final AJE edited version, please visit our verification page. If you have any questions or concerns about this edited document, please contact American Journal Experts at support@aje.com.

American Journal Experts provides a range of editing, translation and manuscript services for researchers and publishers around the world. Our top-quality PhD editors are all native English speakers from America's top universities. Our editors come from nearly every research field and possess the highest qualifications to edit research manuscripts written by non-native English speakers. For more information about our company, services and partner discounts, please visit www.aje.com.

---

## Author Response (AR2)

Author response:

The authors would like to thank Reviewer for comments that improved our manuscript. All comments and suggestions were accepted and changes in the manuscript were done accordingly. Please find our responds bellow.

1. Page 13, line 8 (chap. 3.3.1): BIAS of mslp decreases for elevated stations? The absolute value of the BIAS is higher for elevated stations (Fig. 7), therefore the sentence is misleading. Further, Fig. 7 indicates that some biases of mslp in mountain stations reach 350 hPa. Really? It need some explanation (probably due to different altitudes?).

Author respond:  Careful re-examination of all data is conducted and it is established that at some particular mountain stations observed *mslp* were not reduced to the sea level. We thank the Reviewer for pointing on this mistake. The Fig. 7 and the following text in the manuscript is changed accordingly. The corrections are:

- on the page 13, line 5 - 8

On mountain stations the spread of $\text{BIAS}\left(\overline{(\text{mslp})}_d\right)$ is higher in respect to lower altitudes (-5 to 2.6 hPa for WRF-Chem and -4.9 to 4 hPa for EMEP model), The $\text{BIAS}\left(\overline{(\text{mslp})}_d\right)$ median is the same as for elevated stations for WRF-Chem model and 0.5 hPa for EMEP model.

- on the page 13, line 14 - 15

The models did not show any substantial unsystematic ans systematic errors for $\overline{(\text{mslp})}_d$.

- on the page 17, line 27 - 28

According to the low systematic errors a very good model performance is found in simulating $\overline{(\text{mslp})}_d$ over sea-level and elevated stations, while moderate performance due to high spread of $\text{BIAS}$ over mountain stations.

2. 14/23 (3.3.1): Saying "The overall performance of the models regrading PM10 was good" is not appropriate in my opinion in the perspective of very low correlation with observed data.

Author respond: Based on this comment and comment from Referee 1 in previous Review (the same issue), we decided to avoid clarification of modeling performance in terms of "good - not good", therefore this part is removed (page 14, line 20).

3. Sounding measurements: In chap. 3.3.3 (16/34) there is written time-step of 12 hours, a few row later (17/8) 24 hours. What is correct? Further, Fig. 10 (part Soundings) indicates some given information between 24h steps (e.g. Budapest, around 19.11.) - is there any interpolation used or how is it possible to have a curve between 24h steps?

Author respond: The time step for measurements is 24h, this was corrected (page 13, line 30). There was no specific interpolation for Figure 10. Due to a different number of vertical levels for each timestep up to 500m, the contour plot generated different shapes of filled area. We used `contour` function for plotting black line, and `contourf` for plotting fill in colour between contours of same values. Both functions are part of `Matplotlib` package in Python.

4. 9/2 (2.4.2): The sentence about time-range of model simulations would be better to move in 'common' section 2.4 or write separately by specific models.
Author respond: Comment accepted. The sentence is moved to 2.4 section (page 7, line 13 – 15).

5. 11/35–12/3 (3.2): The explanation of mslp differences in Fig. 3–4 for higher mountain stations is not satisfying. Is it sure that reduction on sea level was performed, eventually performed correctly on these stations? Probably, it is not possible to have such great and highly-located differences in the sea-level pressure field.
Author respond: Following this and the Comment 1, new Figures 3-4 are made and entire paragraph is removed as now observed low *mslp* data are not plotted (e.g., below 900 hPa).

6. 16/25 (3.3.3): Probably Fig. S4 or Fig. 1, not Fig. 4.
Author respond: Can be depicted from either SI Fig S2 or Fig 1. This was corrected in the text (page 16, line 21).

[revised manuscript text omitted]